# Edge-driven nanomembrane-based vertical organic transistors showing a multi-sensing capability

Ali Nawaz [1], Leandro Merces [1], Denise M. de Andrade[1,2], Davi H.S. de Camargo[1,3] & Carlos C. Bof Bufon [1,3 ✉]

The effective utilization of vertical organic transistors in high current density applications demands further reduction of channel length (given by the thickness of the organic semi-conducting layer and typically reported in the 100 nm range) along with the optimization of the source electrode structure. Here we present a viable solution by applying rolled-up metallic nanomembranes as the drain-electrode (which enables the incorporation of few nanometer-thick semiconductor layers) and by lithographically patterning the source-electrode. Our vertical organic transistors operate at ultra-low voltages and demonstrate high current densities (~0.5 A cm$^{-2}$) that are found to depend directly on the number of source edges, provided the source perforation gap is wider than 250 nm. We anticipate that further optimization of device structure can yield higher current densities (~10 A cm$^{-2}$). The use of rolled-up drain-electrode also enables sensing of humidity and light which highlights the potential of these devices to advance next-generation sensing technologies.

[1] Brazilian Nanotechnology National Laboratory (LNNano), Brazilian Center for Research in Energy and Materials (CNPEM), 13083-970 Campinas, São Paulo, Brazil. [2] Department of Materials Engineering, Ponta Grossa State University (UEPG), 84030-900 Ponta Grossa, Paraná, Brazil. [3] Postgraduate Program in Materials Science and Technology (POSMAT), São Paulo State University (UNESP), 17033-360 Bauru, São Paulo, Brazil. ✉email: cesar.bof@lnnano.cnpem.br

Conventional fabrication routes of vertical organic field-effect transistors (VOFETs) involve vertical stacking of a diode cell on top of a capacitor unit. Such a geometry allows the fabrication of devices with nanoscale active channels—given by the thickness of the organic semiconductor (OSC) layer—having a large cross-sectional area. This assists in achieving higher $J_D$ at lower operating voltages[1–4] as compared to the traditional planar OFETs[5,6]. As a result, these devices are useful in applications that demand high current densities, such as light-emitting diode (LED) displays. In addition, VOFETs can be integrated with a photodetector or LED, as an integrated optoelectronic vertical transistor, which further simplifies circuitry production[7–9].

The optimal functionality of a VOFET is dependent on the effective modulation of conductance of the OSC channel. One of the viable methods to achieve this goal involves manipulations in the spatial structure of the source-electrode. In particular, the formation of a perforated source electrode is targeted, which lessens the electrode screening effect and allows the gate field to penetrate through the source perforations into the OSC layer. Furthermore, this enables the accumulation of charge carriers at the dielectric/OSC interface, followed by drift to the drain electrode[10]. In previous reports, the preparation of perforated source has mostly been achieved through the formation of self-assembled electrodes[10–17]. On the other hand, the utilization of photolithographic patterning has rarely been considered[1,8,18]. The VOFET fabrication via photolithography targets high batch-to-batch reproducibility and compatibility with industrial manufacturing routes. Besides, the possibility of a deterministic patterning improves control over the size/shape and density of the source perforations.

One of the primary objectives of fabricating OFETs in a vertical architecture has been the downscaling of transistor channel lengths to obtain higher current densities[3,17]. However, the main limitation for further shrinkage of channel length below 100 nm is related to the deposition of the top drain electrode using conventional evaporation techniques. The deposition of metal through evaporation techniques can cause severe damage to the morphology of the underlying OSC layer and even form pinholes which can result in a short circuit between the source and drain electrodes[19]. Because of such reasons, researchers are usually compelled to use channel lengths that are greater than or equal to 100 nm. However, a viable and attractive way to address this issue involves the preparation of the top drain-electrode using rolled-up metallic nanomembranes (NMs). NMs are nanometer-thick, free-standing structures having typical lateral dimensions in the microscale[20,21]. Sensors[22], solar cells[23], planar OFETs[24], radio-frequency transformers[25], and hybrid organic–inorganic heterojunctions[26,27] are some examples of innovative electronic devices that have benefited from the NM technology. The practically intriguing feature of NM-based devices arises from the possibility of forming bendable, foldable, or even twistable structures, without damaging the surface where it rests. In the case of two-terminal devices, this property has been effectively utilized to promote reliable electrical contacts between the rolled-up NMs and a few nanometer-thick films[26–30], without damaging the morphology of the active layer and further eliminating the limitations imposed by metal deposition[19].

Here, we present the development of a VOFET platform in which the devices are processed entirely via microfabrication techniques and photolithography-assisted patterning. The cylindrically shaped rolled-up metallic NMs play the role of drain electrodes of the VOFETs, enabling the incorporation of sub-50 nm thick OSC (copper phthalocyanine, CuPc) layers. This configuration corresponds to one of the shortest channel lengths

utilized in VOFETs. The controlled fabrication processes reported here result in significantly high current density values (at ultra-low operating voltages), provide saturation in the output characteristics and enable the preparation of compact VOFET structures. Furthermore, the use of rolled-up NM as the top drain electrode enables direct interaction between the OSC layer and the target analyte/light, which leads to the multisensing capability of these devices. To investigate the operating mechanism of the VOFETs, we have used finite-element theoretical simulations, which take into account the current density distributions within the active cell of the VOFET structures. These results reveal the important relationship between the size of the source perforation gap and the dependence of current density distribution on the number of lateral facets (edges) of the source metal. This highlights the role of source electrode edges as a relevant figure-of-merit to deterministically design the source electrode and drastically improve the $J_D$ of both academically and industrially manufactured patterned-source VOFETs. Based on our results, we predict that further optimization of the spatial geometry of source-electrode can yield current densities of ~10 A cm$^{-2}$.

## Results

**Device microfabrication**. In this work, we present a fully integrative on-chip fabrication approach that follows the premises of standard microelectronics, namely, low-cost processing and batch fabrication[31]. The source electrode is carefully designed with a periodic structure that consists of identical circular- or rectangular-shaped perforations (Fig. 1a, b). The device schematic illustration is shown in Fig. 1c, while a confocal laser scanning (CLS) microscopy image of a fabricated device is shown in Fig. 1d. In total, the preparation of rolled-up NM-based VOFETs comprises of eight photolithography and eight thin-film deposition steps. The device preparation steps, together with the respective optical microscopy images are shown in Fig. 2a. In the first step, a *mesa* structure is defined onto which a thin layer of Cr is deposited. The Cr layer also serves as the transistor gate. Sequentially, the Cr-unmasked surface regions (consisting of SiO$_2$) are selectively etched away, and a thin film of Al$_2$O$_3$ is deposited over the entire chip using atomic layer deposition (ALD). The quality of the Al$_2$O$_3$ films was evaluated using ellipsometry measurements, which provided a refractive index of $n = 1.62 \pm 0.02$ (as previously reported for Al$_2$O$_3$ grown by ALD[32,33]). In the second microfabrication step, both the source electrode and contact region are defined, followed by the deposition of Au and SiO$_2$. In the third step, the wet etching of Al$_2$O$_3$ is performed from the top of the gate pad in order to allow contact with the gate. In the fourth step, the drain pad is formed by the deposition of Ag, and the same photolithography step is utilized to deposit more metal onto the gate pad to ensure a good electrical contact during electrical characterization. In the fifth step, a sacrificial layer (Ge) is deposited on top of the active device area, followed by the deposition of an anchor layer (Cr), which also connects the sacrificial layer to the drain pad (step 6). In the second last step, a strained layer, comprising of Au/Ti/Cr NMs, is deposited on top of the sacrificial and anchor layers, followed by the deposition of the CuPc active layer. Finally, the rolling up of the strained metallic NMs is performed in H$_2$O/H$_2$O$_2$ solution. The rolling-up process is assisted by the selective removal of GeO$_x$, which is formed by the oxidation of Ge in the peroxide solution. The metallic NMs roll up into a tubular shape until reaching the anchor layer, at which point they sit elegantly on top of the device active area forming a soft contact with the active layer. Figure 2b shows the photograph of an as-fabricated (9 mm × 9 mm) microchip consisting of 40 VOFETs.

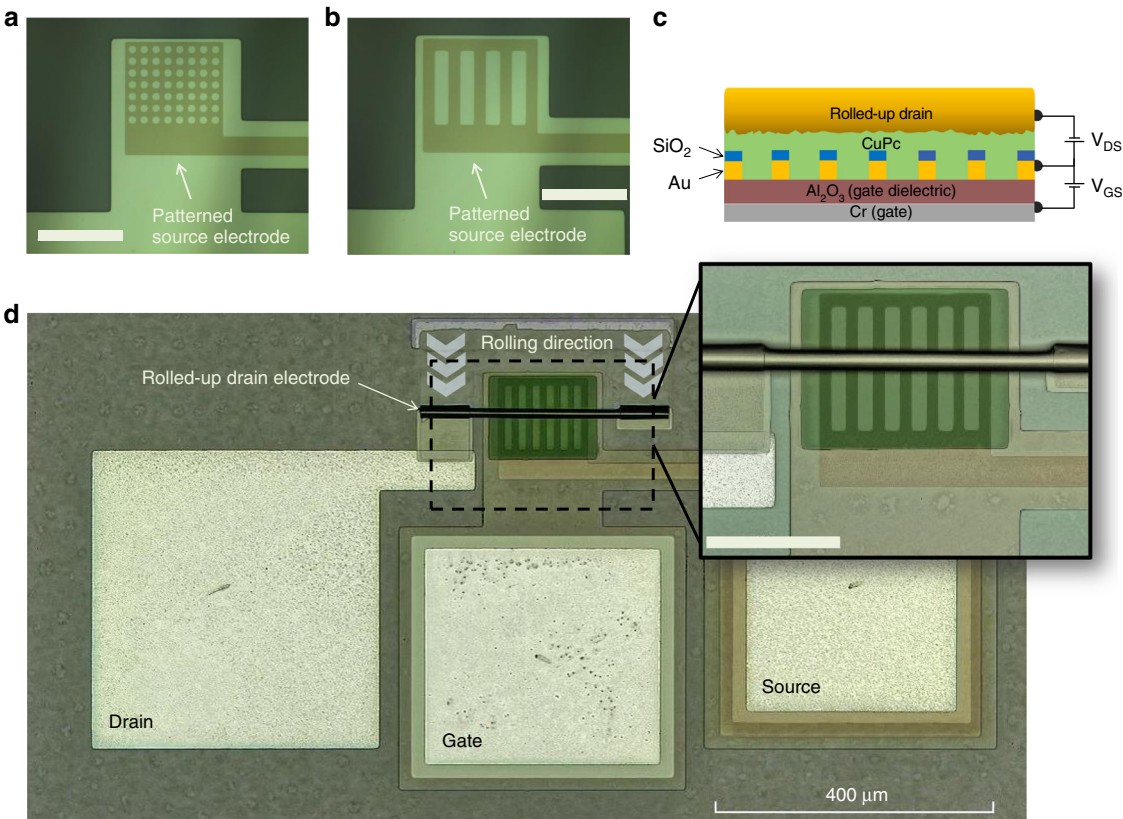

**Fig. 1 Optical and CLS microscopy images along with the VOFET schematic structure. a** Circular and **b** rectangular perforations of the patterned source electrodes captured using an optical microscope (scale bars correspond to 100 μm). **c** Schematic cross-section (front-view) of rolled-up NM-based VOFET. **d** CLS microscopy image of a fully fabricated device. The inset is the zoom in of the active VOFET region showing the rolled-up drain electrode, CuPc film in green and patterned source electrode (scale bar corresponds to 100 μm). The CLS microscopy image is a superposition of an optical image (colored) and a 2D monochromatic image, performed by the software of the Keyence VK-X200 3D laser scanning microscope, which acquires both images in the same measurement. The monochromatic image corresponds to laser scanning (laser wavelength of 408 nm).

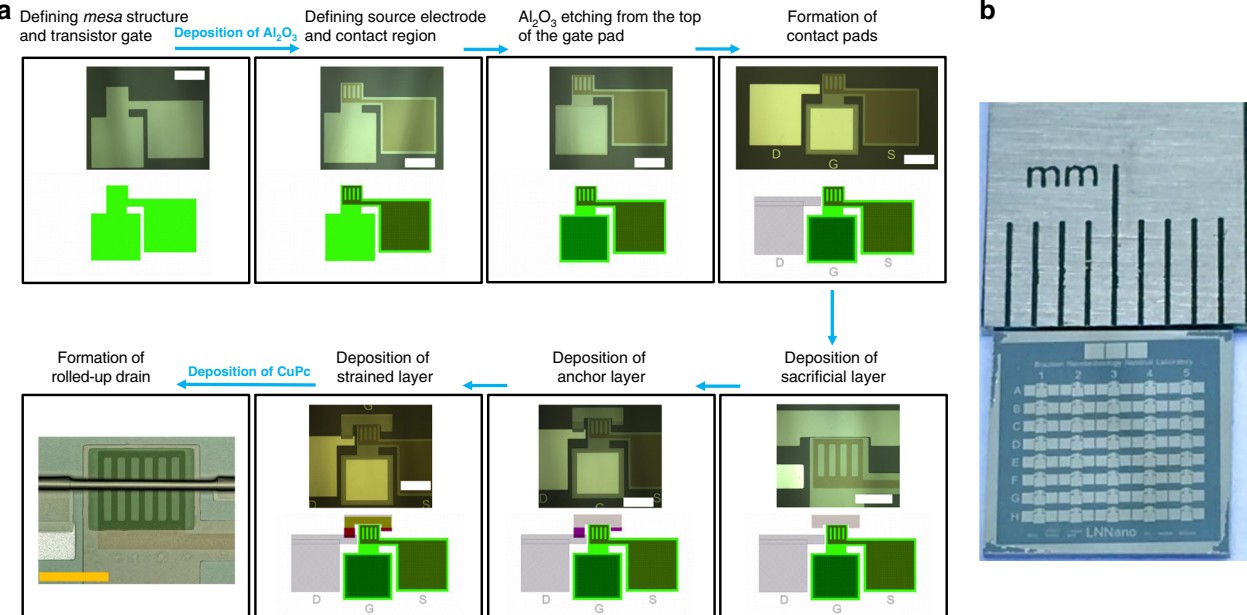

**Fig. 2 VOFET microfabrication steps and a fully fabricated microchip. a** The optical microscopy images taken at the end of every step of the microfabrication process of VOFET devices, together with the photolithography masks designed using the computer-aided program (CleWin). The last panel shows the top-view of the active region of an as-fabricated VOFET. The scale bars in the fifth and eighth panel correspond to 100 μm, whereas, in the rest of the panels the scale bar corresponds to 200 μm. **b** Photograph of a fully fabricated microchip consisting of 40 rolled-up NM-based VOFETs.

**The contact area between rolled-up drain and OSC.** The effective contact area between the rolled-up tube electrode and the CuPc active layer is crucial to the performance of the rolled-up NM-based VOFETs since it determines the effective injecting area. We have estimated the device geometrical contact areas ($A_{geo}$) using the Hertzian model, which takes into account the contact between two elastic half-spaces: a cylinder (the tube electrode) and a horizontal surface (the VOFET base structure; Fig. 3)[34]. Our rolled-up NM-based VOFET structure satisfies all critical assumptions of the Hertzian model, and therefore the

following equations can be used to quantify $A_{geo}$[34]

$$a = \frac{(R\delta)^{1/2}}{2}, \quad (1)$$

$$A_{geo} = aW, \quad (2)$$

where $a$ is the length of the tube electrode contacting the VOFET base structure (see Fig. 3), $R$ is the radius of the tube electrode (4 μm), and $\delta$ is related to the compression of the tube diameter. Since no external compression was applied on the tube electrodes, $\delta$ is the intrinsic tube diameter compression caused by the accommodation of the tube over the base structure. In such a case, $\delta$ is approximately the thickness of the base structure (Fig. 3)[29,35]. Finally, $W$ is the width of the source electrode.

It has previously been demonstrated that, in devices prepared using tube-like electrodes, $A_{geo}$ does not represent the exact electrical contact area ($A_{elect}$)[35,36]. In fact, in the case of nanoscale junctions, $A_{elect}$ has been reported up to 4 orders of magnitude smaller than $A_{geo}$ due to the electrode topography[30,37,38]. In this work, the current densities ($J_D$) are calculated by assuming $A_{elect}$ to be 2 orders of magnitude smaller than $A_{geo}$, which is well within the previously reported limits.

**Experimental results.** Figure 4a, b shows the transfer and output characteristics of VOFET devices in which the source electrode was prepared with identical circular perforations, as shown in the

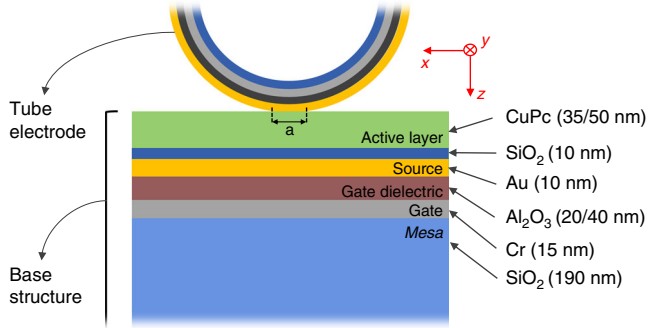

**Fig. 3** Mechanical contact between the rolled-up drain and VOFET base structure.

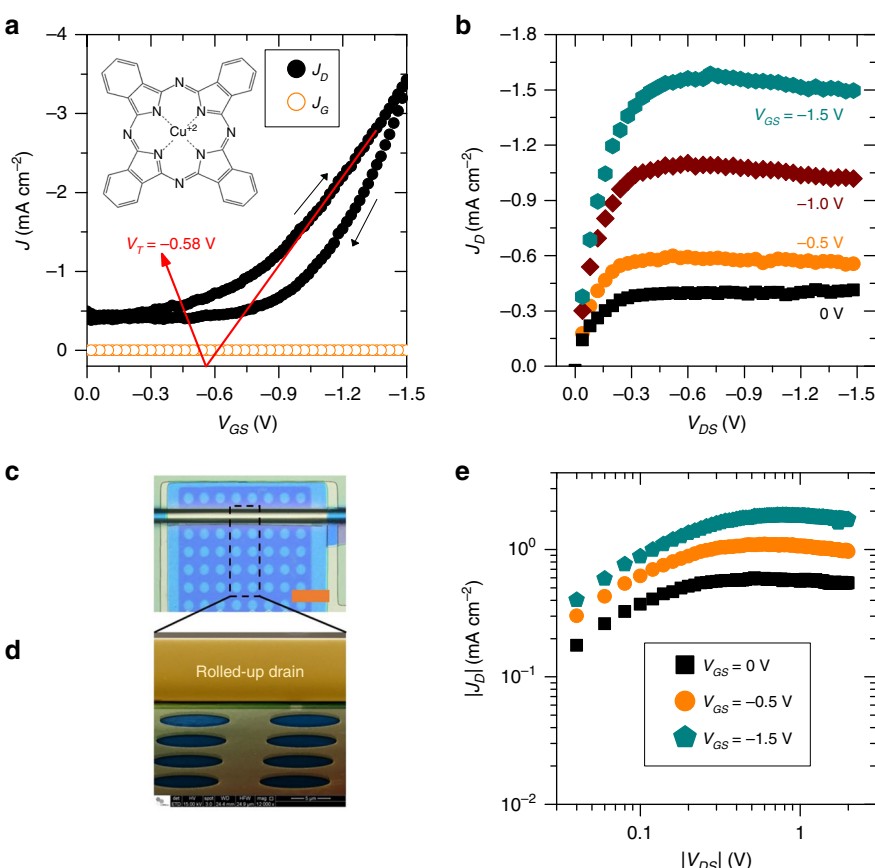

**Fig. 4 NM-based VOFETs prepared with circularly perforated source-electrode. a** $J$ vs. $V_{GS}$ ($V_{DS} = -1.5$ V), and an inset that shows the chemical structure of CuPc. The arrows denote the hysteresis direction. **b** $J_D$ vs. $V_{DS}$. **c** CLS image of the VOFET active region consisting of the rolled-up drain electrode, CuPc film (in blue), and the patterned source electrode with circular perforations (scale bar corresponds to 24 μm). **d** SEM image showing circular perforations of the patterned source, and the rolled-up drain electrode (scale bar corresponds to 5 μm). **e** $J_D$ vs. $V_{DS}$ curves plotted in a log–log scale show the deviation from the power-law dependence of ~2, implying the absence of charge transport paths in the OSC bulk. The transistor dimensions and effective contact area of these devices are: $t_{CuPc} = 50$ nm, $a = 5.43 \times 10^{-5}$ cm, $W = 0.012$ cm, $A_{geo} \approx 6.5 \times 10^{-7}$ cm$^2$, and $A_{elect} \approx 6.5 \times 10^{-9}$ cm$^2$.

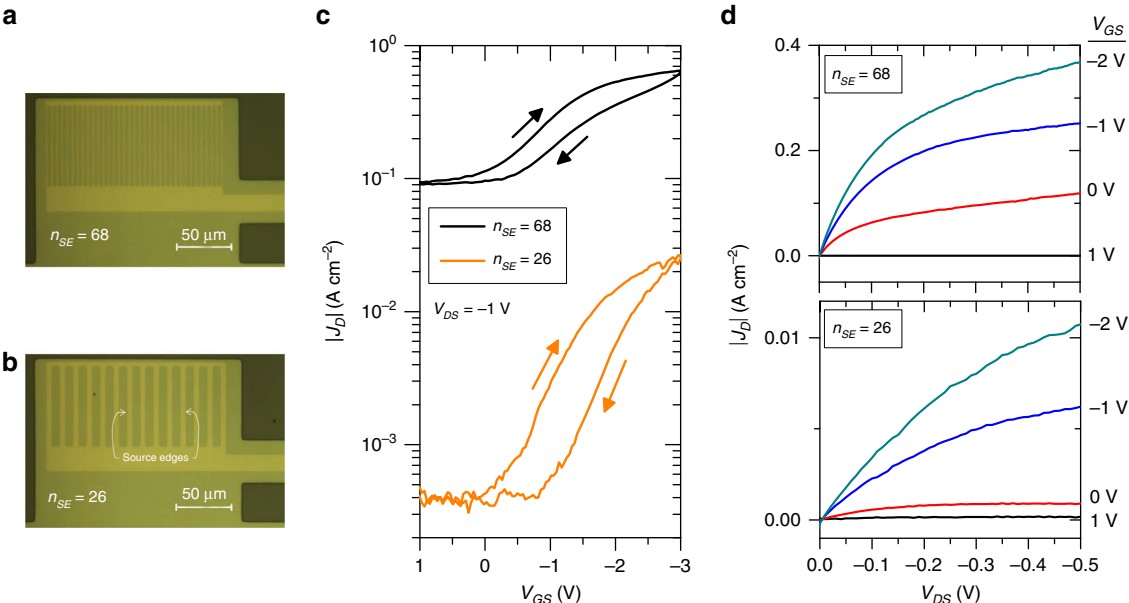

**Fig. 5 NM-based VOFETs consisting of rectangular perforations in the source electrode. a** Optical microscopy image of patterned source electrode having 68 source edges and **b** 26 source edges. **c** $\log|J_D|$ vs. $V_{GS}$ curves and **d** $|J_D|$ vs. $V_{DS}$ curves of VOFETs with 68 and 26 source edges. The arrows denote the hysteresis direction. The transistor dimensions and effective contact area of these devices are $t_{CuPc} = 35$ nm, $a = 5.3 \times 10^{-5}$ cm, $W = 0.024$ cm, $A_{geo} \approx 1.3 \times 10^{-6}$ cm$^2$, and $A_{elect} \approx 1.3 \times 10^{-8}$ cm$^2$.

optical microscopy image (Fig. 4c) and the scanning electron microscopy (SEM) image (Fig. 4d). In this case, the thickness of the CuPc layer is $t_{CuPc} = 50$ nm, whereas, the other important parameters related to device dimensions are $a = 5.43 \times 10^{-5}$ cm, $W = 0.012$ cm, $A_{geo} \approx 6.5 \times 10^{-7}$ cm$^2$, and $A_{elect} \approx 6.5 \times 10^{-9}$ cm$^2$. It must be noted that the gate-current density ($J_G$) was calculated using $A_{geo}$. The VOFETs operate at ultra-low voltages—due to the high dielectric constant of Al$_2$O$_3$, ($\kappa \sim 9$)[39]—and demonstrate a typical transistor behavior with distinguishable on and off states (Fig. 4a). The $J_D$ vs. $V_{GS}$ characteristics of these devices are presented on a logarithmic scale in Supplementary Fig. 1. Even though there is a small energy barrier between the source metal (Au) and the OSC (CuPc), very low off currents ($I_{off}$, i.e., $I_D$ at $V_{GS} = 0$ V) of ~3 pA was recorded. The low values obtained for $I_{off}$ are ensured by depositing a thin insulating layer (composed of SiO$_2$ = 10 nm) on top of the source metal (as illustrated in Fig. 1c), which partially shields the source electrode from the drain potential. The energy diagram of the VOFET layers is shown in Supplementary Fig. 2.

VOFET devices are typically known to demonstrate an absence of saturation in the output characteristics owing to their short channel lengths[2,15,16,40]. In principle, due to the short distance between the source and drain electrodes, the application of $V_{DS}$ leads to large source drain electric fields. This results in the formation of direct transport paths in the bulk of the OSC, that originate from all edges of the source metal and can provide a significant contribution to the drain current, inhibiting the saturation behavior even at high drain voltages. These currents are typically demonstrated by a power-law dependence of ~2 on $V_{DS}$[8,41]. The existence of saturation is considered to be of great importance for applications in which transistors are used as current sources. For instance, AMOLED displays require transistors to reach drain saturation regime in order to maintain a fixed current (brightness) even if a higher voltage is supplied to the OLED pixel[4]. Although some efforts have previously been made to obtain saturation in the output characteristics of VOFET devices[8,18], the adopted strategies increase the overall process

complexity and the risk of damaging the device active region. In this work, we have obtained saturation in the $J_D$ vs. $V_{DS}$ characteristics (Fig. 4b) by covering the top source facets with SiO$_2$ in a single fabrication step. By insulating the top facets, only the lateral parts of the source are exposed to the drain potential, which restricts the contribution of drain currents arising from the bulk of the OSC and leads to the saturation of $J_D$ at high drain voltages. The absence of bulk currents can also be evidenced in the $J_D$ vs. $V_{DS}$ curves plotted in a log–log scale (Fig. 4e) that exhibit deviation from the power law of ~2.

The effective operation of VOFETs as three-terminal devices requires their dielectric cell to demonstrate low leakage current ($I_G$) compared to the output current ($I_D$). The $I_G$ of VOFETs presented in this work was observed to be at least one order of magnitude lower than their $I_D$. This is attributed to the use of the Al$_2$O$_3$ gate dielectric, which effectively suppresses the leakage current between the source and gate electrodes. The low leakage current is also manifested by the overlap of $J_D$ vs. $V_{DS}$ curves at $V_{DS} = 0$ V (Fig. 4b). Another characteristic observed in the $J_D$ vs. $V_{DS}$ characteristics is the slight decrease in $J_D$ at higher $V_{DS}$ values, indicating the presence of trapping sites in the active channel. The same effect also reveals itself in the form of hysteresis in the $J_D$ vs. $V_{GS}$ characteristics (Fig. 4a). We attribute both behaviors to the unavoidable traces of water adsorbed during the microfabrication or characterization of VOFETs, which may result in the formation of water-related charge trapping sites in the device active layer[35,42,43]. In order to reduce the damaging effects caused by the adsorption of water, we stored our samples in a vacuum for at least 3 days before the electrical measurements are performed.

One of the current limitations of our VOFET platform is the low on–off current ratio ($I_{on}/I_{off}$) of ~10 in the above-mentioned devices. This performance is associated with the very small $A_{elect}$, which leads to a relatively low $I_{on}$. In order to improve these values, the width of the source electrode was increased to $W = 0.024$ cm (such that $A_{elect} \approx 1.3 \times 10^{-8}$ cm$^2$). At the same time, $t_{CuPc}$ was further decreased to 35 nm. As shown in Fig. 5a, b, the

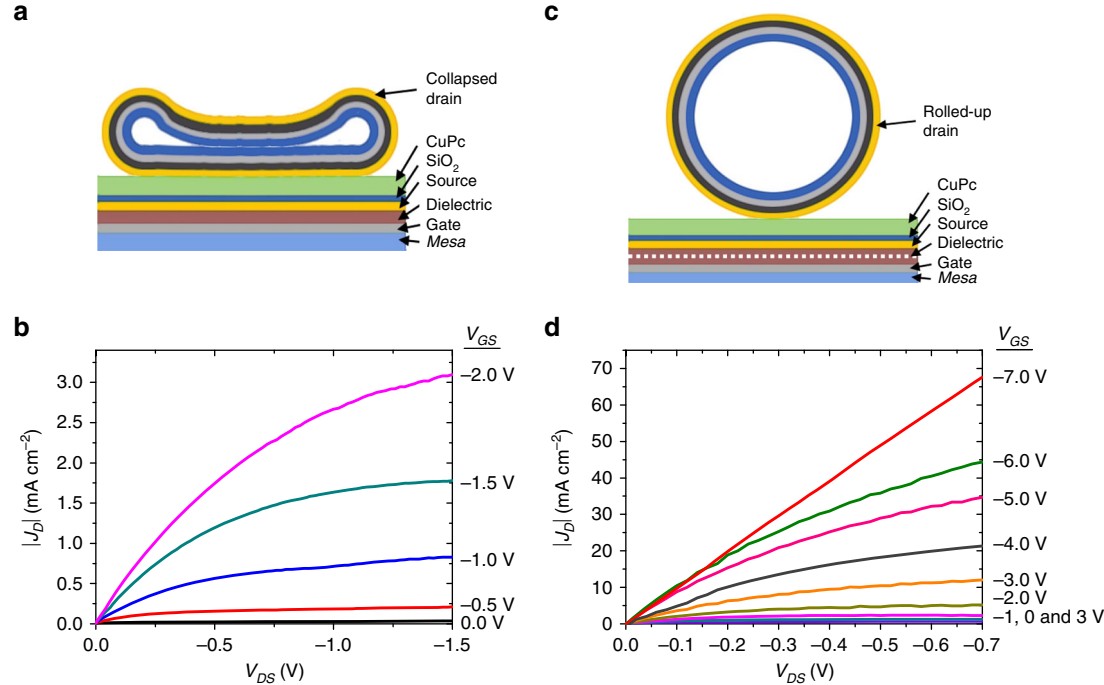

**Fig. 6 Irreversible compression over the rolled-up drain and thicker gate dielectric layer. a** The cross-sectional view of the collapsed tube electrode formed by the application of an irreversible compression over the tube electrode. **b** $|J_D|$ vs. $V_{DS}$ of VOFET in which an irreversible compression over the tubular drain electrode was applied. Here, the VOFET consisted of 26 source edges, while $t_{CuPc} = 35$ nm, $W = 0.024$ cm, $A_{geo} \approx 3 \times 10^{-5}$ cm$^2$, and $A_{elect} \approx 3 \times 10^{-7}$ cm$^2$. **c** The cross-sectional view of rolled-up NM-based VOFET structure in which a 40 nm Al$_2$O$_3$ layer is incorporated. The dotted line indicates the top position of the dielectric layer prior to increasing its thickness. **d** $|J_D|$ vs. $V_{DS}$ of rolled-up NM-based VOFET in which 40 nm Al$_2$O$_3$ layer was utilized. In this case, the transistor also consisted of 26 source edges, while $t_{CuPc} = 35$ nm, $a = 5.5 \times 10^{-5}$ cm, $W = 0.024$ cm, $A_{geo} \approx 1.3 \times 10^{-6}$ cm$^2$, and $A_{elect} \approx 1.3 \times 10^{-8}$ cm$^2$.

source electrode was, in a novel configuration, patterned with identical rectangular perforations. The change in source patterns from circular to rectangular perforations was principally brought about in order to ensure that the rolled-up drain tube always overlaps with the perforated region of the source electrode. The representative electrical results are shown in Fig. 5c, d. As expected, the devices demonstrated significant improvement in $J_D$. In principle, $J_D$ was found to be dependent on the number of source edges ($n_{SE}$). Source edges are defined as the lateral facets of source metals that are orthogonal to the axis of the rolled-up drain electrode, as indicated in Fig. 5b (two edges surrounding a single source perforation). When transistors were fabricated using $n_{SE} = 26$, the devices showed a drain-current density of ~10 mA cm$^{-2}$, while the devices with $n_{SE} = 68$ showed a considerable increase in the drain-current density to ~500 mA cm$^{-2}$. In both cases, $I_{on}/I_{off}$ was observed to be within the range of ~40. One of the intriguing features of these devices is the direct dependence of $J_D$ on the number of source edges. This point will be further discussed in the theoretical modeling section.

In an attempt to further improve $I_{on}/I_{off}$, two additional techniques were performed in devices having $n_{SE} = 26$. In the first case, an irreversible compression over the tubular drain electrode was applied, thus modifying its shape from cylindrical to a folded sheet (illustrated in Fig. 6a). The objective was to increase the drain/semiconductor contacting area and as a consequence $I_{on}/I_{off}$. The irreversible compression of the tube electrode is induced by the capillary effect when the samples are dried fastly during the rolling-up process. This results in collapsing the metallic NM that takes the form of a flat surface instead of a cylindrical shell. The optical microscopy image of a collapsed tube electrode is provided in Supplementary Fig. 3. The geometrical contact area

of a collapsed system can be calculated using the following expression

$$A_{geo} = \pi R W, \qquad (3)$$

where $\pi R$ is half of the tube circumference. When the tube collapses, we consider that the circumference becomes a straight line with the bottom facet of the tube (circumference/2) in contact with the semiconductor. For $R = 4$ μm and $W = 240$ μm, the geometrical contact area becomes $A_{geo} = 3 \times 10^{-5}$ cm$^2$, whereas, the electrical contact area, $A_{elect} = 3 \times 10^{-7}$ cm$^2$. The output characteristics of a collapsed tube VOFET are shown in Fig. 6b. Compared to the cylindrical tube VOFETs, the collapsed tube devices demonstrated an improvement in $I_{on}/I_{off}$ by one order of magnitude. In this case, $J_D$ at $V_{GS} = -2$ V was recorded as ~3 mA cm$^{-2}$, while $J_D$ at $V_{GS} = 0$ V was recorded as ~0.03 mA cm$^{-2}$ (Fig. 6b), yielding $I_{on}/I_{off}$ of ~10$^2$. The second technique used to improve the $I_{on}/I_{off}$ of the VOFETs involved the utilization of a thicker gate dielectric layer of 40 nm (illustrated in Fig. 6c). By improving the dielectric strength, the devices could be operated at higher voltages (Fig. 6d). Here, the devices demonstrated on- and off-current densities of ~66 mA cm$^{-2}$ and ~0.65 mA cm$^{-2}$, respectively. The $I_{on}/I_{off}$ of ~10$^2$ is one order of magnitude higher than the one for 20 nm Al$_2$O$_3$ VOFETs (Fig. 5d).

The use of rolled-up NM as the top drain electrode presents an opportunity for the direct interaction between the OSC layer and external chemical/physical analytes or light. By taking this feature to our advantage, we tested the response of the VOFET devices after exposure to humidity and light (Fig. 7). In the case of humidity sensing, the devices showed a significant decrease in $J_D$ when relative humidity levels were increased from 60 to 75%

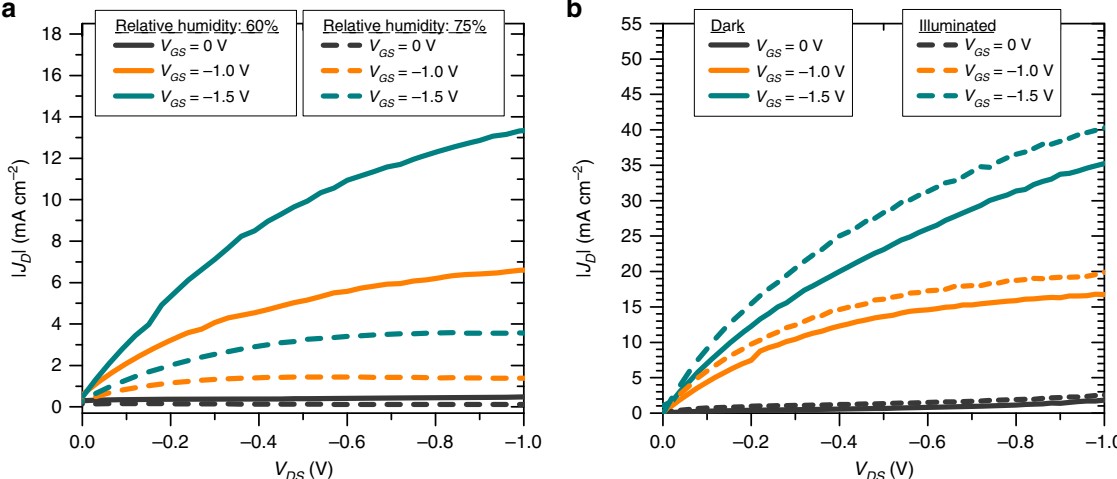

**Fig. 7 Multi-sensing capability of the rolled-up NM-based VOFETs. a** $|J_D|$ vs. $V_{DS}$ of rolled-up NM-based VOFET performed at different humidity levels. **b** $|J_D|$ vs. $V_{DS}$ of rolled-up NM-based VOFET performed in dark and illuminated environments. The optical power of the illumination source is 150 W while the power per unit area was calibrated at 100 mW cm$^{-2}$. In both panels, the transistors consisted of 26 source edges, while $t_{CuPc} = 35$ nm, $a = 5.5 \times 10^{-5}$ cm, $W = 0.024$ cm, $A_{geo} \approx 1.3 \times 10^{-6}$ cm$^2$, and $A_{elect} \approx 1.3 \times 10^{-8}$ cm$^2$.

(Fig. 7a). This indicates that the adsorption of water molecules on the CuPc layer results in the possible formation of charge traps in the transistor interface regions or in the OSC bulk. For instance, the H$_2$O-related traps would hinder the injection/extraction of charge carriers if they are formed at the dielectric/OSC interface within the perforated regions of the patterned source electrode or near the rolled-up NM regions. A more precise explanation would, of course, require a detailed study based on elaborate electrical/physical characterization results. However, the objective of the present work is to demonstrate the capability of these VOFETs to work as humidity sensors because the use of rolled-up NM as the top drain electrode allows a large area of the OSC to be exposed to the target analyte molecules.

The operation of the rolled-up NM-based VOFETs as phototransistor was tested by illuminating the devices using a solar simulator operating at an optical power of 150 W (Fig. 7b). A slight increase in $J_D$ can be observed when the device is illuminated. This can be attributed to the formation of photogenerated electron–hole pairs in the active VOFET channel following the absorption of photons by the CuPc layer. Photoresponsivity ($R$) is considered to be an important parameter to characterize the performance of phototransistors. It is expressed as[44]

$$R = \frac{\Delta I_D}{PA}, \quad (4)$$

where $\Delta I_D$ is the photocurrent generated under illumination, $P$ is the illumination intensity in the unit area, and $A$ is the active device area. The specific detectivity ($D^*$) is another important parameter that determines the capability of a phototransistor to respond to a weak light signal. Assuming that the shot noise from the dark current is the major contributor to the total noise, $D^*$ is expressed as[44]

$$D^* = \frac{RA^{1/2}}{\left(2eI_{D,dark}\right)^{1/2}}, \quad (5)$$

where $e$ is the electron charge in coulombs, and $I_{D,dark}$ is the dark current. The $R$ and $D^*$ of the devices presented in Fig. 7b were measured to be ~0.05 A W$^{-1}$ and ~10$^9$ Jones, respectively, at $V_{DS} = -1$ V and $V_{GS} = -1.5$ V. With our experiments, we have demonstrated the strong potential of rolled-up NM-based

VOFETs to work as phototransistors. The photoresponse of our devices can be further improved by optimizing some fabrication parameters. For instance, the incorporation of photoactive materials, such as lead sulfide quantum dots[45], and 9,10-diphenylanthracene single-crystal[46], into the device architecture can substantially improve the phototransistor performance.

**Theoretical modeling**. The theoretical simulations were performed in COMSOL Multiphysics. As in the experiments, the VOFET device structures comprise six layers stacked one upon the other. Herein, only one device active cell was considered (2D cross-section, front-view shown in Fig. 8a), which spatially considers a single source perforation, surrounded by the metallic source layer (defining two source edges). The simulations were performed for three different perforation gaps (50 nm, 200 nm, and 3 μm) while the devices were biased at $V_{GS} = V_{DS} = -2$ V. Figure 8b–d (upper panels) shows the current distributions (denoted by arrows) within the device active cell, while the color gradient corresponds to the electric potential. The lower panels of Fig. 8b–d show the normalized current density distribution as a function of the size of each perforation gap, extracted from the respective simulation results. Because of the incorporation of an insulating layer over the top facet of the source metal, the injection of charge carriers from the source electrode occurs solely from its lateral facets, while the vertical conducting channel is formed as a result of the electric fields induced by the gate and drain potentials.

In the case of a 50 nm perforation gap (Fig. 8b), the current density distribution is concentrated at the center of the entire perforation gap (as evident in the lower panel), where the effective vertical channel forms a "tunnel-like" shape. Essentially, the lateral electric fields, projected by the source edges, inject the charge carriers into the perforation gap. On the other hand, the vertical electric fields projected by the gate and drain electrodes, are opposite in direction and turn the current to the vertical pathway. Hence, a vertical channel is formed in which the current flows within a "tunnel" from the center of the perforation gap to the drain electrode. In such devices, the operating mechanism relies on the vertical "tunnel" channel[10]. The formation of this "tunnel" can be considered as a limitation to device performance since it forces the current to flow in a very narrow portion of the

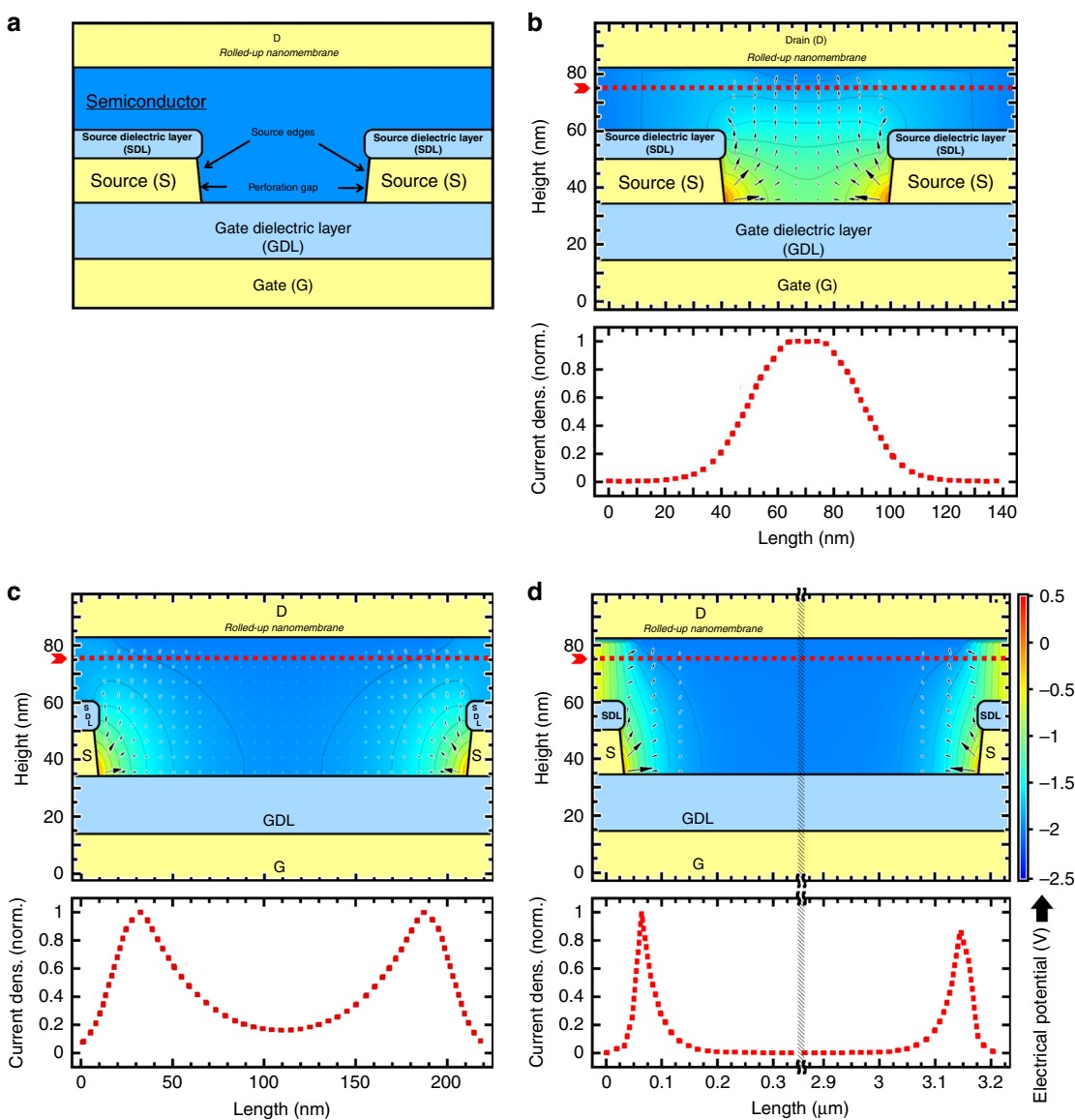

**Fig. 8 Theoretical simulations of current density distributions. a** Simulation 2D cross-section (front view) of the device active cell with the layers notations. Panels **b**–**d** show the finite-element simulations accounting for the distributions of current density and electric potential within the active cell of the VOFET structures with ON-state biasing conditions ($V_{DS} = V_{GS} = -2$ V, the source is grounded). The simulations were performed by considering source electrodes with **b** 50 nm, **c** 200 nm, and **d** 3 μm perforation gaps. For all the architectures, the solutions are plotted along with the OSC layer. The current density distributions are indicated with arrows, while the electric potential can be visualized by the color-scale (provided at the right-hand side). The solid lines are equipotential. For each perforation set, the vertical components of the current density at the vicinity of the drain—along with the profile dotted-lines—are exhibited at the bottom of panels (**b**–**d**), respectively.

perforation gap, thereby, limiting the output current densities. Therefore, in order to further improve the current density of patterned source VOFETs, it is required to devise a method that allows the current to flow independently of the vertical "tunnel". The formation of a "tunnel" basically depends on the biasing conditions and device structural parameters such as source electrode thickness, dielectric thickness, and the size of the perforation gap[10]. The idea of the present theoretical study is to succinctly demonstrate that, while keeping the biasing conditions and rest of the structural parameters constant, a subtle departure from the "tunnel" effect can be achieved by only varying the size of the perforation gap. In this context, the attention of the reader is brought toward Fig. 8c, in which a 200 nm source perforation gap was considered. It can be seen that the "tunnel" structure is starting to fade away, and the current density distribution is no longer concentrated at the center of the perforation gap (as

evident in the lower panel of Fig. 8c). In the case of a 3 μm perforation gap, the current densities are completely concentrated in the vicinities of the source edges, and the vertical channel no longer forms the "tunnel" structure. A close inspection of the lower panels of Fig. 8b–d shows that the minimum size of the perforation gap required to depart entirely from the "tunnel" effect is ~250 nm. This evidences that, upon reaching the threshold size of the perforation gap (~250 nm), the current density that forms the conducting channel starts to depend on the source edges rather than being dependent on the vertical "tunnel" formed at the center of the perforation gap. This supplements the current understanding of patterned source VOFETs[10] and further implies that one of the most critical parameters to obtain an increase in the current density is the number of source edges divided by the source width. This observation was also demonstrated in our experimental results where micrometer-

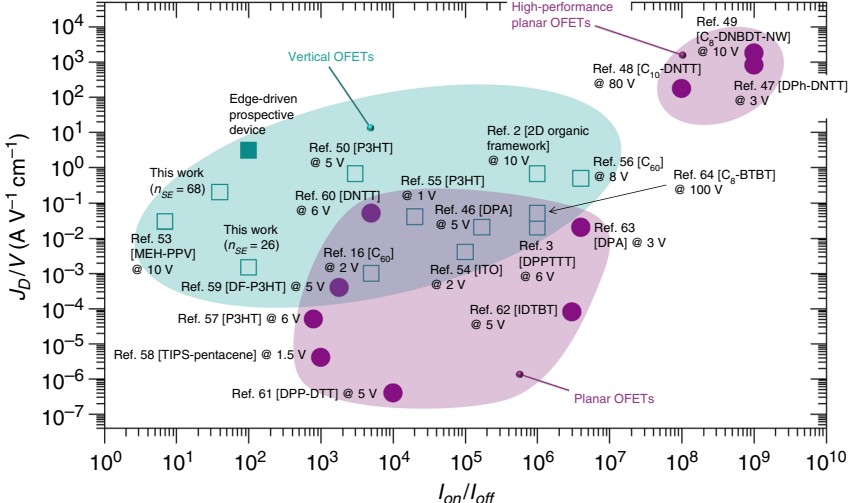

**Fig. 9 Summary of the performance of rolled-up NM-based VOFETs.** The graph shows a comparison of the $J_D$ (divided by $V_{GS}$) and $I_{on}/I_{off}$ obtained in this work with various vertical and planar OFETs reported in the literature[2,3,16,46–50,56–67]. The respective operation voltages ($V_{GS}$) are also indicated. The optimization of edge-driven VOFETs can result in at least a tenfold increase in $J_D$ values. The solid teal symbol at the top of the graph indicates the $J_D$ and $I_{on}/I_{off}$ of a prospective device in which the patterned source electrode is 240 μm wide and comprises of 960 source edges. The $I_{on}/I_{off}$ of the prospective device is estimated considering that the transistor utilizes a compressed (collapsed) nanomembrane electrode.

sized perforation gaps were utilized to fabricate the VOFETs. We showed that an increase in the number of source edges from 26 to 68 (while keeping the source width constant) resulted in an increase in the current density by a factor of ~40 (Fig. 5c).

**Practical applicability and future perspectives.** The current densities observed in this work are comparable to several state-of-the-art OFET and VOFET devices (Fig. 9). We attribute this to the use of thin OSC layers (<50 nm) and the deterministic patterning of the source electrode. The performance of our devices is still not comparable to the high-performance OFETs obtained by sophisticated device engineering[47] or prepared using 2D OSC crystals[48,49]. Nevertheless, the current densities obtained in this work at $V_{GS} \leq 3$ V are comparable to what was recently obtained for 2D organic framework/graphene-based VOFETs operating at $V_{GS} = 10$ V[2], and single crystal-based VOFETs operating at $V_{GS} = 5$ V[46]. In fact, most of the high-performance devices (Fig. 9) operate at $V_{GS} \geq 10$ V[48,49], which raises concerns regarding the integration of these devices in next-generation applications such as flexible OLED displays. The ultra-low voltage operation is a remarkable feature of the VOFETs presented in this work. Whereas most of the benchmark VOFET devices operate at $V_{GS} \geq 5$ V[2,3,18,46,48–50], our devices demonstrate comparable or even higher $J_D$ values at relatively smaller operating voltages ($V_{GS} \leq 3$ V). This also complies with one of the fundamental ideas behind the development of organic transistors in a vertical architecture, which is to enable high current densities at low operating voltages such that practical (portable) applications in organic electronics can be ensured[3,51].

According to the theoretical simulations, the dependence of current density on source edges can be maintained by utilizing perforation gaps as narrow as ~250 nm. If such an arrangement is applied to a patterned source electrode having a width of 240 μm, it will yield 960 source edges. For comparison, our devices comprised of 26 and 68 source edges (when perforation gaps of 9 μm and 3 μm, respectively were utilized) for patterned source electrodes having a width of 240 μm (Fig. 5a, b). In our VOFET results, we observed that the increase in the number of source edges from 26 to 68 results in an increase of $J_D$ from ~10 to ~500 mA cm$^{-2}$. By assuming a linear relationship between

source edges and $J_D$, we anticipate that, for a case where 960 source edges are utilized, $J_D \approx 10$ A cm$^{-2}$ (implying, $J_D/V_{GS} \approx 3.3$ A V$^{-1}$ cm$^{-2}$) can be achieved (Fig. 9). The fabrication of a VOFET device having a significantly increased number of source edges would, however, require the use of nano-fabrication tools to obtain precise control over the spatial geometry of the source electrode. Here, we would like to stress that the source edge-driven mechanism is not limited to our rolled-NM devices. In fact, it can also be used for further performance optimization of patterned-source VOFETs reported in literature[8,13,18,52], to expedite the commercialization of next-generation electronic applications like LED displays.

Although the $I_{on}/I_{off}$ of the present platform is relatively low, however, as demonstrated by our sensor results, one of the remarkable features of these devices is their multisensing capability. Hence, we stress that our device platform is not only a transistor-alone platform but brings with it the great promise of using these VOFETs as transduction elements in innovative sensor technologies. In this context, the rolled-up NM-based devices can also be considered for the development of gas-/biosensors, pressure sensors, and memory devices[53]. For instance, the inherent flexibility and mechanical robustness of NMs provide the opportunity to deform them by applying an external compression controllably. As a result, the interface between the NM electrode and the underlying OSC layer can be manipulated to control the current injection area, and thus the output current density. Such a behavior directly indicates the potential applicability of rolled-up NM-based VOFETs as compression gauges. A similar sensing concept was recently demonstrated by Merces et al.[30] in variable-area transport junctions.

**Discussion**

We have demonstrated the development of a VOFET platform in which the device preparation relies entirely on photolithography patterning while the top drain electrode is formed using rolled-up metallic NMs. The use of rolled-up drain electrode has enabled the incorporation of very thin OSC layers (35 nm), corresponding to one of the shortest channels ever reported in VOFET devices. In these devices, the source electrode is carefully patterned with a periodic structure that consists of identical circular- or

rectangular-shaped perforations. This configuration allows gate field to effectively penetrate the OSC layer and result in current densities of ~0.5 A cm$^{-2}$. Furthermore, the VOFETs demonstrate saturation in the output characteristics, which is obtained by covering the top facets of the source with a thin insulating layer that restricts the contribution of drain currents arising from the bulk of the OSC. It is observed that the current density scales with the number of source edges, which is confirmed through theoretical simulations. The calculations show that upon reaching a specific threshold size of the perforation gap, the current density starts to depend on the source edges. This highlights the role of source edges for further enhancement of current densities in patterned-source VOFETs. Based on our results, we anticipate that further optimization of the spatial structure of the patterned source electrode can yield current densities of ~10 A cm$^{-2}$. The edge-driven operating mechanism extends the current understanding of VOFETs, and the important point is that it can be applied to drastically improve the $J_D$ of both academically and industrially manufactured patterned-source devices. Similarly, the use of rolled-up NMs as the top drain electrode highlights the potential of incorporating ultra-short OSC channels (<10 nm) which would also assist in further enhancement of the output current densities. Finally, we demonstrated that the rolled-up NM-based VOFETs show clear variations in output currents when exposed to humidity and light, thus highlighting the multisensing capability of these devices and their strong potential in innovative sensor technologies.

## Methods

**VOFET microfabrication**. The VOFET devices were fabricated on 9 × 9 mm Si/SiO$_2$ substrates, which were sequentially cleaned in an ultrasonic bath in acetone and isopropanol, followed by O$_2$ plasma for 5 min. Before the first fabrication step, the substrates were passivated by spin-coating hexamethyldisilazane (HMDS, obtained from Technic, Inc.) at 3000 rpm for 30 s, followed by a soft bake for 3 min at 100 °C. A reverse-mode lithography step defined the gate electrode and contact regions after which Cr (15 nm) was deposited by electron-beam evaporation at a rate of ~0.5 Å s$^{-1}$ (base pressure of ~10$^{-7}$ Torr). A lift off was performed to remove the lithography mask and conclude the formation of the gate electrode and a contact region. Sequentially, the *mesa* structure was prepared by reactive ion etching of 190 nm SiO$_2$ using the Oxford PLASMAPro 80 etching system. In this case, tetrafluoromethane (CF$_4$) was used as the etching gas while a previously deposited Cr layer was used as the etching mask. The gate dielectric layer (Al$_2$O$_3$: 20 nm or 40 nm) was grown using ALD at 150 °C with trimethylaluminum (TMA; Sigma-Aldrich) and water as precursors. This deposition was performed on a Cambridge NanoTech Savannah 100 ALD system. The refractive index of the Al$_2$O$_3$ thin films was measured using the ellipsometry technique, wherein, a light source having a wavelength of 633 nm was utilized. Assuming the high reproducibility of Al$_2$O$_3$ films grown by ALD, we expect a transmittance of ca. 90–98% considering visible light. The source electrode and contact pad region were defined by a reverse-lithography step, after which, Cr, Au, and SiO$_2$ (5, 10, and 10 nm, respectively) were deposited in sequence by electron-beam evaporation at a rate of ~0.5 Å s$^{-1}$ (base pressure of ~10$^{-7}$ Torr). A lift-off was performed to remove the lithography mask then concluding the formation of the source electrode and contact pad region. The width of the source electrode was either 120 μm (for circular patterned source) or 240 μm (for rectangular patterned source). The etching of Al$_2$O$_3$ from the top of the gate contact pad was performed by firstly protecting the other parts of the substrate using the photoresist (AZ 5214E). In this case, direct-mode lithography was performed, and sequentially Al$_2$O$_3$ was etched with hydrogen fluoride (HF) aqueous solution (1% v/v) for 18 s. In the next step, reverse-mode lithography was performed in order to define the contact pad regions. Sequentially, Cr and Ag (10 and 70 nm) were deposited by electron-beam evaporation at a rate of ~0.5 Å s$^{-1}$ (base pressure of ~10$^{-7}$ Torr), followed by lift off to conclude the formation of contact pads. It must be noted that the lift-off processes were always performed using acetone.

The sacrificial layer was formed of GeO$_x$. Firstly, Ge (20 nm) was deposited using e-beam evaporation, whereas, the oxidation of the Ge layer was performed later in the H$_2$O$_2$ solution during the rolling-up process. After the formation of the sacrificial layer, an anchor layer of Cr (10 nm) was deposited sequentially using e-beam evaporation. In the next step, a strained layer composed of Au (5 nm), Ti (15 nm), Cr (20 nm), and SiO$_2$ (10 nm) was deposited. The Ti/Cr/SiO$_2$ trilayer was used to generate the strain gradient necessary for rolling, whereas Au was used as the layer contacting CuPc and the drain pad after the roll up. In sequence, CuPc (35 or 50 nm) was deposited using Leybold UNIVEX 250 at a rate of ~0.2 Å s$^{-1}$. The patterning of the sacrificial layer, anchor layer, strained layer, and CuPc layer

was performed via reverse-mode lithography. After the deposition of the CuPc layer, the roll-up of the strained layer was performed in H$_2$O:H$_2$O$_2$ (1:0.0025) solution, facilitated by the self-release of the strained layer through selective etching of the GeO$_x$ sacrificial layer. The total roll-up time took around 15 min, and the diameter of the rolled-up NM was measured as ~8 μm using CLS microscopy. In some cases, an irreversible compression of the tube electrode was induced by lifting the samples quickly from the peroxide samples. This results in faster drying of the samples, leading to the collapse of the metallic NM that takes the form of a flat surface instead of a cylindrical shell. The optical microscopy image of a collapsed tube electrode is provided in Supplementary Fig. 3.

**Photolithography process**. In all the photolithography steps, an image reversal photoresist (AZ 5214E purchased from Microchemicals GmbH) was used, the mask alignment and UV exposure were performed on a direct-write photolithography machine (MicroWriter ML3), and the development of lithography masks was performed using AZ 726 MIF photoresist stripper (obtained from Merck).

**Electrical and sensor characterization**. The VOFETs electrical characteristics were evaluated by using a Keithley 4200 SCS coupled to an MPS150 Cascade Microtech probe station. The samples were stored in vacuum for at least 3 days before performing the electrical measurements. For the VOFET characterization in different humidity conditions, we have used a homemade acrylic chamber prepared on the sample stage of the MPS150 probe station. Hence, controlled N$_2$ flow was employed to carry H$_2$O vapor through the chamber. The humidity levels were recorded by a Minipa MTH-1362 hygrometer coupled to the system. The phototransistor characterization was performed using an Oriel Instruments solar simulator that was coupled to the MPS150 probe station. The light beam was simulated by a xenon arc lamp having an optical power of 150 W. The power per unit area was calibrated with a radiometer at 100 mW cm$^{-2}$.

**Physical characterization**. The optical, CLS, and SEM images were taken on a Zeiss Axio Imager.A2 microscope, Keyence VK-X200 3D laser scanning microscope, and Inspect F50 scanning electron microscope, respectively. The coloring of different VOFET layers in the SEM image was performed using the Paint.NET software.

**Theoretical modeling**. The finite-element simulations were performed in COMSOL Multiphysics. A 2-dimensional space coupled to the quasi-statics electric mode was employed to simulate in-plane electric currents in the VOFET structures. The calculations were done along a representative device cross-section composed of gate, gate dielectric, source, OSC and drain (playing the role of the rolled-up NM) —from the bottom to the top of the VOFET, respectively. The component materials were defined similarly to the as-fabricated devices—i.e., Al$_2$O$_3$ was used to compose the dielectric layers, CuPc was used as the OSC, and Au was used as material for the gate, source, and drain electrodes. The electrical properties, such as conductivity ($\sigma$) and relative permittivity ($\varepsilon$) of both, dielectric layers and electrodes, were given by the software library. For the OSC layer, we have used $\sigma_{CuPc}$ ~10$^{-3}$ S m$^{-1}$ [54], and $\varepsilon_{CuPc}$ = 4.5[55]. The interfacial resistance of the source and drain electrodes were adjusted to recover the typical on-current distribution of the VOFET with a nanometer-sized source perforation gap. To complete the simulation, the terminal voltages were selected accordingly to the experimental data obtained during the device operation, i.e., $V_{GS} = V_{DS} = -2$ V, with the source electrode being grounded. The same parameters were used to analyze the picture of the source edges on the device current distribution, finally considering a micrometer-sized perforation gap in comparison to the nanometer-sized one. A discussion accounting for both current distribution and electric potential at the devices is proposed to elucidate the edge-limited behavior exhibited by the rolled-up NM-based VOFETs.

## Data availability

The authors declare that all relevant data supporting the findings of this work are available from the corresponding author on request.

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

## Acknowledgements

The authors acknowledge LNNano/CNPEM for advanced infrastructure and technical support. The authors also acknowledge FAPESP (14/25979-2, 16/25346-5, and 18/18136-0), CNPq (408770/2018-0), CAPES and National Institute for Complex Functional Materials (INCT) for financial support. C.C.B.B. is a productivity research fellow from CNPq (305305/2016-6). C.C.B.B also acknowledges the support of CNPq and FAPESP (Brazil) through Inomat, INCT (CNPq Proc 465452/2014-0 and FAPESP 14/50906-9). Professor A.A. Quivy's group is acknowledged for the evaluation of refractive indices using the ellipsometry technique, and M.P. Pereira is acknowledged for her support with the confocal laser scanning microscopy. This article is dedicated to the memory of late Prof. Dr. Ivo Alexandre Hümmelgen, who greatly contributed to the field of organic electronic devices, and significantly advanced the research area of vertical transistors in Brazil.

## Author contributions

A.N. executed the project, fabricated the devices, characterized and extracted the results, and wrote the paper. D.M.A. and D.H.S.C. provided the technical support in the microfabrication processes. L.M. was the main contributor to the theoretical simulations while he also contributed to discussions and paper revision. C.C.B.B. designed the device's concept and experiments, supervised the project, and provided fruitful discussions and contributed to the revision of the paper.

## Competing interests

The authors declare no competing interests.
