## [Peer Review File · Nature Communications]

Reviewers' comments:

Reviewer #1 (Remarks to the Author):

The authors demonstrated a vertical organic field-effect transistor (VOFET) devices developed in this manuscript with high current densities (1 A/cm^2) at very low operating voltages ($< 1.5\text{V}$). This was possible since the devices have very thin organic semiconductor (OSC) active layer ($35 \text{ nm}/50 \text{ nm}$). Making VOFET with very thin organic active layer is a challenge because the organic layer could be damaged during the metal electrode evaporation process. The authors circumvent this issue by introducing rolled-up metallic nanomembranes as the top drain-electrode and thus, they could achieve the VOFET with ultra-thin OSC active layer. In addition, the devices demonstrate saturated output characteristics. The typical VOFETs do not have the drain saturation regime and this is one of the limitations of VOFETs. The authors covered the top facets of the source with an insulating layer (SiO_2) to prevent OSC bulk current and thereby the VOFETs with saturated output was achieved.

The novelty is to introduce rolled-up metallic nanomembranes as the top drain-electrode to fabricate the VOFETs with very thin OSC active layer. Developing the simple process (depositing a thin insulating layer on top of the source electrode) for making VOFET with saturated output characteristics.

One of the main ideas of this manuscript is to use metallic nanomembrane as the top drain-electrode. Since this method does not cause damage to the underlying organic layer, it looks a very good approach to achieve devices with a few nanometer-thick organic layers. However, the devices inherently have very small electrode contact area because of the wire morphology of the rolled-up metallic membrane. So the drain current of the devices is very small. As a result, the devices show very low on-off ratio ($\sim 10^2$). It seems like the authors achieved the thin organic layer VOFET at the expense of on-off ratio which is one of the major parameters of the transistor. The authors claimed that LED could be the potential application of their devices.

However, the calculated on-off ratio of the device is very low ($\sim 10^3$). I don't see the device can be improved and the authors use a very complicated method to fabricate a vertical transistor.

Reviewer #2 (Remarks to the Author):

In their manuscript "Edge-driven vertical organic field-effect transistors with rolled-up drain electrode," Ali Nawaz and coworkers report on the fabrication of vertical organic field-effect transistors in which the drain contact has the shape of a hollow metallic cylinder created by the controlled release of a mechanically strained nanomembrane from the substrate surface, allowing it to roll onto itself to form a rolled-up nanotube with a diameter of a few microns that comes to rest on the surface of the organic semiconductor layer. The transistors were fabricated on a silicon substrate by sequential deposition and photolithographic patterning of a Cr gate electrode, a 20-nm-thick aluminum oxide gate dielectric (deposited by atomic layer deposition), a deliberately perforated gold source contact (covered with a thin silicon-dioxide blocking layer deposited by electron-beam evaporation), a vacuum-deposited organic semiconductor layer (copper phthalocyanine), and a rolled-up Au nanotube as the drain contact.

To calculate the density of the drain current flowing through their transistors, the authors divide the measured drain current (which reaches a maximum of about 1 nA) by the fraction of the area through which this current is believed to flow from the organic semiconductor into the drain contact. For a nanotube-shaped drain contact with a diameter of 8 microns, this area is assumed to be $6.5\text{e-}9 \text{ cm}^2$ for a contact width of 120 microns, and $1.3\text{e-}8 \text{ cm}^2$ for a width of 240 microns. Based on these assumptions, the authors report a maximum on-state drain-current density (at

gate-source and drain-source voltages of -1.5 V) of 12 mA/cm². Depending on the transistor design, the off-state drain-current density (measured at a gate-source voltage of zero and a drain-source voltage of -1.5 V) ranges from 0.4 to 1.5 mA/cm². The maximum on/off ratio is 8. (These are the values I see in the output curves shown in the manuscript; the authors claim an on-state drain-current density of 1 A/cm² and an on/off ratio of 100 on page 11 and an on/off ratio of 200 in Figure 7.)

I agree that the idea to use a rolled-up nanotube as a drain contact in organic field-effect transistors is unique and intriguing, so despite the fact that I do not share the authors' enthusiasm regarding the prospects of this concept for useful devices, I recommend publication in Nature Communications, provided the following comments and questions are adequately addressed in a revised manuscript:

On page 5, the authors mention that the use of photolithography and thin-film deposition "allows several components to be fabricated in parallel." This implies that the use of other fabrication methods would not allow components to be fabricated in parallel. Please give a few relevant examples of such methods.

In Figure 2b, the authors compare their organic-transistor chip to an individually packaged MOSFET. What are the authors trying to imply or convey with this comparison? Are they suggesting that their fabrication approach allows a larger transistor-integration density than conventional industrial transistor-fabrication techniques? In this case, I recommend a comparison with the Apple A12X or the Samsung Exynos 9825.

On page 6, the authors claim that the drain contact plays "the role of effective injecting area." This is not true, since the charge carriers are injected from the source and extracted into the drain (hence the terms "source" and "drain"), i.e., the drain is responsible for carrier extraction, not carrier injection.

In Figure 4a, the authors use the symbol "J" for the drain-current density, whereas in Figures 4b, 4e, 5c, 5d and 7, they use the symbol "JD". Why are different symbols used for the same parameter?

In Figure 4a, the authors plot a parameter called "JG", which is presumably the gate-current density. The authors need to explain in the text to which area the measured gate current was normalized to calculate this parameter.

For gate-source and drain-source voltages of -1.5 V, the authors show on-state drain-current densities of 3.5 mA/cm² in Figure 4a (transfer curve) and 1.5 mA/cm² in Figure 4b (output curve), for presumably the same transistor. What is the reason for this discrepancy?

In Figure 4e, the authors report a drain-current density of 0.02 mA/cm², which is two orders of magnitude smaller than the drain-current density reported in Figures 4a and 4b for presumably the same transistor. Is the reason for this discrepancy that the drain-current density in Figure 4e was calculated using the geometrical contact area (6.5e-7 cm²), rather than the electrical contact area (6.5e-9 cm²)?

On pages 9 and 10 and in the captions to Figures 4 and 5, the authors use the symbol "x" to denote a parameter being measured and plotted as a function of another parameter. However, the correct terminology to denote this type of measurement or plot is "versus" or "vs." or "as a function of", whereas the symbol "x" is usually used to denote the product of two parameters.

On page 11, the authors mention an on-state drain-current density of 1 A/cm² and an on/off ratio of 200, but these values are not supported by the measurement data shown in Figure 5d, where the on-state drain-current density (at gate-source and drain-source voltages of -1.5 V) is 12

mA/cm², the off-state drain-current density (at a gate-source voltage of zero and a drain-source voltage of -1.5 V) is 1.5 mA/cm², and hence the on/off ratio is 8. Perhaps the authors have extracted some or all of the values from the transfer curves shown in Figures 4a and 5c, but given the extreme non-idealities of these transfer curves, they should not be trusted. To be honest, the transfer curves look nothing like what would be expected from a field-effect transistor. I strongly recommend that the authors only give values extracted from the output curves, not from the transfer curves.

On page 14 and in Figure 7, the authors benchmark the on-state drain-current density and the on/off ratio of their vertical organic transistors against those of vertical and planar organic transistors reported in the literature. I enjoy benchmarking, and I like the idea of Figure 7, but I request that additional data points from the following three recent publications on planar organic transistors are added to Figure 7: Yamamura et al. (*Science Advances*, vol. 4, p. eaao5758, 2018), Borchert et al. (*Nature Communications*, vol. 10, p. 1119, 2019), Zhou et al. (*Advanced Science*, vol. 6, p. 1900775, 2019). The transistors reported in these publications have on/off ratios of 1e8 to 1e9 and drain currents between about 0.1 and 2.5 mA. The current density flowing through the semiconductor layer of these planar transistors can be calculated by dividing the drain current by the cross-sectional area of the semiconductor layer, i.e., by the product of the channel width and the semiconductor thickness, which yields current densities between 10 and 40 kA/cm² (0.5 to 4 kA/Vcm² when normalized to the gate-source voltage). These data points must be shown in Figure 7.

For the “prospective” transistor shown in Figure 7, the authors indicate an on/off ratio of 1000. The authors need to explain how this value was determined.

The transistor dimensions and the values of the electrical contact area (A_{elect}) should be given in the figure captions to Figure 4 and Figure 5.

In addition to Figure 4a and Figure 5c, please also show graphs in which the drain current or drain-current density is plotted on a logarithmic scale (as opposed to a linear scale) as a function of the gate-source voltage.

References 8, 35, 55, 58 and 67 are missing the volume and page numbers.

Reviewer #3 (Remarks to the Author):

A. Nawaz et al. developed an innovative vertical organic field-effect transistor (OFET) with a rolled-up drain-electrode which improves the effective device construction with the high current densities at ultra-low operating voltage. Combined with the theoretic study, the author also demonstrated the thickness dependent carrier transport mechanism in the VOFETs, which predicates the strong potential towards obtaining higher current density > 10 A/cm² with further architecture optimization. Even though the manuscript analyzed with lots of experimental results and simulation, there are several problems need to be solved before further consideration.

(1) The author strength the high device performance with the CuPc based VOFETs, which display a current density of 1 A/cm² but a limited current on/off ratio. However, there are literatures reported excellent electric properties with current density > 1 A/cm² and high on/off ratio > 10⁶ based on organic semiconductors, such as C8-BTBT and C60 (*Nat. Commun.* 2014, 5, 5162; *Appl. Phys. Lett.* 2004, 85, 5084.). The author should optimize parameters such as A_{elect} , OSC materials and film thickness, which confirm the effective enhancement of on current.

(2) For the electric properties characterization, the author attributed the threshold voltage shifts and hysteresis to the absorption of water. As mentioned in this work, the measurements were performed in air, can the author compare the results with detection in N₂ conditions? What's more, there are reports via post-processing to reduce the solvent effect on organic electronic devices (*Nat. Mater.* 2017, 16, 356; *Nat. Commun.* 2019, 10, 2122; *Adv. Mater.* 2018, 30, 1801874.), the

author need introduce post-treatment such as annealing, vacuum storage, to reduce the water effect.

(3) The theoretical research did the series investigation on the thickness (50 nm to 3 μm) and source edges (26 to 68). The interesting thing is that the experiments were performed with the large channel length (9 and 3 μm). How about the device performance and carrier transport of thinner films devices?

(4) The author demonstrated an effective strategy on VOFETs fabrication and the comparable properties to different OSCs (Figure 7). The author still needs to do more literature research to add materials with high JD and $I_{\text{on}}/I_{\text{off}}$ (such as Nat. Commun. 2014, 5, 5162). What's more, can the author show us an application with these CuPc based VOFETs, which possess high current density at low-operation voltage? Such as in the phototransistor or memory elements (Nat. Photonics 2016, 10, 129; Adv. Mater. 2018, 30, 18036557; Adv. Mater. 2017, 29, 1604769; Adv. Funct. Mater. 2019, 29, 1808453.)

Comments of Reviewer # 1 and response by the authors

- *In the italic text*: the reviewer's comments.
 - In the normal text: taken actions.
 - **In bold notation**: major changes incorporated in the manuscript.
-

Reviewer # 1:

The authors demonstrated a vertical organic field-effect transistor (VOFET) devices developed in this manuscript with high current densities (1 A/cm^2) at very low operating voltages ($< 1.5 \text{ V}$). This was possible since the devices have very thin organic semiconductor (OSC) active layer (35 nm/50 nm). Making VOFET with very thin organic active layer is a challenge because the organic layer could be damaged during the metal electrode evaporation process. The authors circumvent this issue by introducing rolled-up metallic nanomembranes as the top drain-electrode and thus, they could achieve the VOFET with ultra-thin OSC active layer. In addition, the devices demonstrate saturated output characteristics. The typical VOFETs do not have the drain saturation regime and this is one of the limitations of VOFETs. The authors covered the top facets of the source with an insulating layer (SiO_2) to prevent OSC bulk current and thereby the VOFETs with saturated output was achieved.

The novelty is to introduce rolled-up metallic nanomembranes as the top drain-electrode to fabricate the VOFETs with very thin OSC active layer.

Developing the simple process (depositing a thin insulating layer on top of the source electrode) for making VOFET with saturated output characteristics.

One of the main ideas of this manuscript is to use metallic nanomembrane as the top drain-electrode. Since this method does not cause damage to the underlying organic layer, it looks a very good approach to achieve devices with a few nanometer-thick organic layers. However, the devices inherently have very small electrode contact area because of the wire morphology of the rolled-up metallic membrane. So the drain current of the devices is very small. As a result, the devices show very low on-off ratio ($\sim 10^2$). It seems like the authors achieved the thin organic layer VOFET at the expense of on-off ratio which is one of the major parameters of the transistor. The authors claimed that LED could be the potential application of their devices.

However, the calculated on-off ratio of the device is very low ($\sim 10^3$). I don't see the device can be improved and the authors use a very complicated method to fabricate a vertical transistor.

Author's response:

We thank the reviewer for the positive evaluation of our manuscript. Beyond the two topics highlighted by the reviewer (*i.e.*, using nanomembrane-based drain electrodes to electrically contact nanometer-thick semiconductors, and developing a reliable process to reach output saturation in microfabricated VOFETs), our contribution also elucidates the working principle of the reported VOFET devices using a theoretical modeling study. By employing a patterned source, we were able to set the current level of our VOFETs by changing the number and/or size of source perforations (keeping constant the source porosity). Accordingly, our experiments showed that in VOFETs fabricated with micrometer-wide source perforations, the output current density is not dependent on the source porosity but on the number of source edges. Our theoretical analysis corroborated our experimental findings by demonstrating that the current density profiles along the semiconducting channel have their shapes modified when the source perforation size changes from 50 nm to 3 μm (Figure 8b-d).

Our values of the on-off current ratio are low, however, we should emphasize that our manuscript opens up possibilities of developing and improving the conventional VOFET platform reported in the literature. The nanomembrane-based drain and the edge-driven features both extend the concept of conventional VOFET operation. In fact, the source-edge-based working mechanism can also be employed in conventional VOFET systems (fabricated using evaporated drain electrode) to obtain ultra-high current densities and on-off current ratio.

Indeed, without any post-processing treatment, we obtain relatively low on-off current ratios by employing a conventional organic semiconductor (CuPc). However, with the comments of the reviewer in mind, we performed two additional techniques in order to achieve considerable improvement in the on-off current ratios of our VOFETs:

- (i) We have applied an irreversible compression over the tubular drain electrode, thus modifying its shape from cylindrical to a folded sheet. The irreversible compression was introduced by collapsing the tube electrode during the rolling-up process. As a consequence, the drain/semiconductor contacting area increased, as well as the on-off current ratio (to $\sim 10^2$). The new output characteristics are shown in Figure 6a, and the corresponding text has been inserted/modified on page 12 (highlighted in yellow). We have also included an optical microscopy image of a collapsed tube electrode as Supplementary Figure 2.
- (ii) We have increased the thickness of the Al_2O_3 gate dielectric layer (from 20 nm to 40 nm) to possibly tune its dielectric strength. As a consequence, we have obtained samples that support gate voltages up to 7 V. Under such conditions, the on-off current ratio also showed significant improvement to $\sim 10^2$. The new output characteristics are shown in Figure 6b, and the corresponding text has been inserted/modified on page 12 (highlighted in yellow).

In addition to the above two experiments, in our revised manuscript we have also demonstrated the use of our VOFETs as humidity and light-sensitive units. The use of rolled-up nanomembrane as the top drain electrode allows direct interaction between the semiconducting layer and external chemical analytes and light. Hence, evident changes in the magnitude and shape of the output characteristics were observed when the devices were exposed to incident light and increased relative humidity. The corresponding output characteristics are presented in the revised manuscript as Figure 7, while the corresponding discussion is inserted on pages 13 and 14. Our new multi-sensing results indicate that rolled-up nanomembrane VOFETs may also be used for the detection of various gases and biomolecules. Hence, our device platform should not only be seen as a transistor-alone platform but as a potential candidate to develop and improve innovative sensing technologies.

Comments of Reviewer # 2 and response by the authors

- *In the italic text*: the reviewer's comments.
 - In the normal text: taken actions.
 - **In bold notation**: significant changes incorporated in the manuscript.
-

Reviewer # 2:

In their manuscript "Edge-driven vertical organic field-effect transistors with rolled-up drain electrode," Ali Nawaz and coworkers report on the fabrication of vertical organic field-effect transistors in which the drain contact has the shape of a hollow metallic cylinder created by the controlled release of a mechanically strained nanomembrane from the substrate surface, allowing it to roll onto itself to form a rolled-up nanotube with a diameter of a few microns that comes to rest on the surface of the organic semiconductor layer. The transistors were fabricated on a silicon substrate by sequential deposition and photolithographic patterning of a Cr gate electrode, a 20-nm-thick aluminum oxide gate dielectric (deposited by atomic layer deposition), a deliberately perforated gold source contact (covered with a thin silicon-dioxide blocking layer deposited by electron-beam evaporation), a vacuum-deposited organic semiconductor layer (copper phthalocyanine), and a rolled-up Au nanotube as the drain contact.

To calculate the density of the drain current flowing through their transistors, the authors divide the measured drain current (which reaches a maximum of about 1 nA) by the fraction of the area through which this current is believed to flow from the organic semiconductor into the drain contact. For a nanotube-shaped drain contact with a diameter of 8 microns, this area is assumed to be $6.5e-9$ cm² for a contact width of 120 microns, and $1.3e-8$ cm² for a width of 240 microns. Based on these assumptions, the authors report a maximum on-state drain-current density (at gate-source and drain-source voltages of -1.5 V) of 12 mA/cm². Depending on the transistor design, the off-state drain-current density (measured at a gate-source voltage of zero and a drain-source voltage of -1.5 V) ranges from 0.4 to 1.5 mA/cm². The maximum on/off ratio is 8. (These are the values I see in the output curves shown in the manuscript; the authors claim an

on-state drain-current density of 1 A/cm² and an on/off ratio of 100 on page 11 and an on/off ratio of 200 in Figure 7.)

I agree that the idea to use a rolled-up nanotube as a drain contact in organic field-effect transistors is unique and intriguing, so despite the fact that I do not share the authors' enthusiasm regarding the prospects of this concept for useful devices, I recommend publication in Nature Communications, provided the following comments and questions are adequately addressed in a revised manuscript:

- 1. On page 5, the authors mention that the use of photolithography and thin-film deposition “allows several components to be fabricated in parallel.” This implies that the use of other fabrication methods would not allow components to be fabricated in parallel. Please give a few relevant examples of such methods.*

Author's response: We thank the reviewer for the confident evaluation of our manuscript. When we mentioned that photolithography and thin-film deposition allow several components to be fabricated in parallel, it is not an implication that the use of other methods would not allow doing so. To avoid such an interpretation, we have improved such a particular discussion in the manuscript (page 5) as follows: **“The preparation of VOFETs presented in this work is entirely based on photolithography patterning and thin-film deposition (8 steps each). This fully integrative on-chip approach follows the premises of standard microelectronics that involve the need for both low-cost processing and batch fabrication”** The updated text is highlighted in yellow in the revised manuscript.

- 2. In Figure 2b, the authors compare their organic-transistor chip to an individually packaged MOSFET. What are the authors trying to imply or convey with this comparison? Are they suggesting that their fabrication approach allows a larger transistor-integration density than conventional industrial transistor-fabrication techniques? In this case, I recommend a comparison with the Apple A12X or the Samsung Exynos 9825.*

Author's response: The purpose of comparing our VOFET microchip with a conventional MOSFET was not to imply that our fabrication approach allows a larger transistor-integration density than conventional industrial transistor-fabrication techniques. In fact, in Figure 2b, we are essentially showing the size of as-fabricated VOFET microchip prepared using

photolithography-assisted patterning and microfabrication techniques available at the Brazilian National Nanotechnology Laboratory. We highlight the miniaturized size of the microchip by comparing it with a conventional MOSFET. To allow a clear interpretation of Figure 2b, we have added the following phrase in the *Device microfabrication* section on page 6: **“To better visualize the overall dimensions of our microchip, a conventional MOSFET is also placed in the photograph, which is almost 20 mm in length.”** This phrase has also been highlighted in yellow in the revised manuscript.

3. *On page 6, the authors claim that the drain contact plays “the role of effective injecting area.” This is not true, since the charge carriers are injected from the source and extracted into the drain (hence the terms “source” and “drain”), i.e., the drain is responsible for carrier extraction, not carrier injection.*

Author’s response: We understand the concern of the reviewer. The corresponding sentence on page 6 has been modified and highlighted in yellow in the revised manuscript.

4. *In Figure 4a, the authors use the symbol “J” for the drain-current density, whereas in Figures 4b, 4e, 5c, 5d and 7, they use the symbol “JD”. Why are different symbols used for the same parameter?*

Author’s response: The symbol “J” corresponds to current density, “J_D” corresponds to drain-current density, and “J_G” corresponds to gate-current density. In Figure 4a, only the symbol “J” is used since this Figure shows both drain-current density (J_D) and gate-current density (J_G). The symbols J_D and J_G are highlighted inside Figure 4a beside the chemical structure of CuPc. In other Figures, when the symbol “J_D” is used, only drain-current density is showed.

5. *In Figure 4a, the authors plot a parameter called “JG”, which is presumably the gate-current density. The authors need to explain in the text to which area the measured gate current was normalized to calculate this parameter.*

Author’s response: We understand the concern of the reviewer. Throughout the manuscript, J_G has been calculated using A_{geo} . This has now been explicitly mentioned in the section “Experimental results” on page 8 (highlighted in yellow).

6. For gate-source and drain-source voltages of -1.5 V , the authors show on-state drain-current densities of 3.5 mA/cm^2 in Figure 4a (transfer curve) and 1.5 mA/cm^2 in Figure 4b (output curve), for presumably the same transistor. What is the reason for this discrepancy?

Author’s response: We thank the reviewer for pointing out the discrepancy. The electrical characteristics presented in Figures 4a and 4b were indeed measured on the same transistor. The slight discrepancy between the on-state drain-current density in the transfer curve (Figure 4a) and the output curve (Figure 4b) is because the transfer curve was measured using the “normal” configuration, while the output curve was measured using the “quiet” configuration of the Keithley Semiconductor Parameter Analyzer. During the measurements performed in the “normal” configuration, it was found that the current takes some time (tens of seconds) to reach the stationary state during the dc voltage sweep (see Figure below the response). On the other hand, in the “quiet” configuration, the parameter analyzer allows the current to reach the stationary state before the dc voltage sweep is initiated. For this reason, the slight discrepancy between the on-state drain-current density in the transfer curve (Figure 4a) and the output curve (Figure 4b) is observed. During the revision of our manuscript, we repeated several experiments and improved our analysis to ensure the operational reliability of our VOFETs. As a consequence, it can be observed that the new transfer and output characteristics of Figures 5c and 5d, measured in the “quite” configuration show similar on-state drain-current densities.

Figure Current as a function of time performed using the “normal” configuration of Keithley Semiconductor Parameter Analyzer. It can be seen that current takes about 20 seconds to reach stationary state during the measurement.

7. In Figure 4e, the authors report a drain-current density of 0.02 mA/cm^2 , which is two orders of magnitude smaller than the drain-current density reported in Figures 4a and 4b for presumably the same transistor. Is the reason for this discrepancy that the drain-current

density in Figure 4e was calculated using the geometrical contact area ($6.5e-7 \text{ cm}^2$), rather than the electrical contact area ($6.5e-9 \text{ cm}^2$)?

Author's response: We thank the reviewer for pointing out the mistake. The drain-current density in Figure 4e was indeed calculated using the geometrical contact area ($6.5 \times 10^{-7} \text{ cm}^2$) instead of the electrical contact area ($6.5 \times 10^{-9} \text{ cm}^2$). Figure 4 has now been updated such that in both Figures 4b and 4e the drain-current density is calculated using the electrical contact area ($6.5 \times 10^{-9} \text{ cm}^2$).

8. *On pages 9 and 10 and in the captions to Figures 4 and 5, the authors use the symbol “x” to denote a parameter being measured and plotted as a function of another parameter. However, the correct terminology to denote this type of measurement or plot is “versus” or “vs.” or “as a function of”, whereas the symbol “x” is usually used to denote the product of two parameters.*

Author's response: We understand the concern of the reviewer and we have now replaced the symbol “x” by “vs.” throughout the manuscript.

9. *On page 11, the authors mention an on-state drain-current density of 1 A/cm^2 and an on/off ratio of 200, but these values are not supported by the measurement data shown in Figure 5d, where the on-state drain-current density (at gate-source and drain-source voltages of -1.5 V) is 12 mA/cm^2 , the off-state drain-current density (at a gate-source voltage of zero and a drain-source voltage of -1.5 V) is 1.5 mA/cm^2 , and hence the on/off ratio is 8. Perhaps the authors have extracted some or all of the values from the transfer curves shown in Figures 4a and 5c, but given the extreme non-idealities of these transfer curves, they should not be trusted. To be honest, the transfer curves look nothing like what would be expected from a field-effect transistor. I strongly recommend that the authors only give values extracted from the output curves, not from the transfer curves.*

Author's response: We understand the concern of the reviewer and following their comment we have taken the opportunity to repeat several of our experiments and further improve our analysis by characterizing more devices. Accordingly, we have realized that some samples were exhibiting current variations during the application of a voltage step. During such variations (that could reach tens of seconds) the output current was not found stationary, giving rise to the

discrepancies between output and transfer characteristics indicated by the reviewer. Notice that output curves are obtained for sweeping the dc voltage between source and drain, while transfer curves keep the source-drain voltage steady during measurement. We could reach reproducibility by measuring the respective devices in the “quite” configuration of the Keithley Semiconductor Parameter Analyzer, which allows the current to reach the stationary state before the measurement is initiated. Hence, we have now improved our manuscript with appropriate data and analysis to ensure the operational reliability of our VOFETs (see Figure 5, 6, 7 and respective discussion in pages 11–14). Following the recommendation of the reviewer, we have now recalculated the J_D and I_{on}/I_{off} values using the output curves.

10. *On page 14 and in Figure 7, the authors benchmark the on-state drain-current density and the on/off ratio of their vertical organic transistors against those of vertical and planar organic transistors reported in the literature. I enjoy benchmarking, and I like the idea of Figure 7, but I request that additional data points from the following three recent publications on planar organic transistors are added to Figure 7: Yamamura et al. (Science Advances, vol. 4, p. eaao5758, 2018), Borchert et al. (Nature Communications, vol. 10, p. 1119, 2019), Zhou et al. (Advanced Science, vol. 6, p. 1900775, 2019). The transistors reported in these publications have on/off ratios of $1e8$ to $1e9$ and drain currents between about 0.1 and 2.5 mA. The current density flowing through the semiconductor layer of these planar transistors can be calculated by dividing the drain current by the cross-sectional area of the semiconductor layer, i.e., by the product of the channel width and the semiconductor thickness, which yields current densities between 10 and 40 kA/cm² (0.5 to 4 kA/Vcm² when normalized to the gate-source voltage). These data points must be shown in Figure 7.*

Author’s response: We thank the reviewer for indicating these important references which have now been included in Figure 9 (of the revised manuscript) as high-performance OFETs. Similarly, the corresponding text in the section “Discussion” has also been updated (highlighted in yellow in the revised manuscript).

11. *For the “prospective” transistor shown in Figure 7, the authors indicate an on/off ratio of 1000. The authors need to explain how this value was determined.*

Author's response: In the revised manuscript, the I_{on}/I_{off} of the prospective device is estimated considering that the transistor utilizes a compressed (collapsed) nanomembrane electrode. For the collapsed system, we measured I_{on}/I_{off} of $\sim 10^2$ (Figure 6a of the revised manuscript) which is maintained as it is for the edge-driven prospective device. The “comparison of J_D/V_{GS} and I_{on}/I_{off} ” figure is now shown in Figure 9 in the revised manuscript. The above text has also been inserted in the “Discussion” section on page 19 (highlighted in yellow).

12. The transistor dimensions and the values of the electrical contact area (A_{elect}) should be given in the figure captions to Figure 4 and Figure 5.

Author's response: The transistor dimensions and the values of A_{geo} and A_{elect} have now been included in the figure captions of all the figures showing device electrical characteristics.

13. In addition to Figure 4a and Figure 5c, please also show graphs in which the drain current or drain-current density is plotted on a logarithmic scale (as opposed to a linear scale) as a function of the gate-source voltage.

Authors' response: Following the suggestion of the reviewer, the logarithmic plot of J_D vs. V_{GS} presented in Figure 4a has been included as Supplementary Figure 1. On the other hand, Figure 5c itself is updated to present the drain-current density in the logarithmic scale.

14. References 8, 35, 55, 58 and 67 are missing the volume and page numbers.

Author's response: We thank the reviewer for pointing out the missing volume and page numbers of the references 8, 35, 55, 58 and 67, which have now been updated. These references are now cited as Ref. 9, 36, 61, 64 and 73, respectively in the revised manuscript.

Comments of Reviewer # 3 and response by the authors

- *In the italic text*: the reviewer's comments.
 - In the normal text: taken actions.
 - **In bold notation**: significant changes incorporated in the manuscript.
-

Reviewer # 3:

A. Nawaz et al. developed an innovative vertical organic field-effect transistor (OFET) with a rolled-up drain-electrode which improves the effective device construction with the high current densities at ultra-low operating voltage. Combined with the theoretic study, the author also demonstrated the thickness dependent carrier transport mechanism in the VOFETs, which predicates the strong potential towards obtaining higher current density $> 10 \text{ A/cm}^2$ with further architecture optimization. Even though the manuscript analyzed with lots of experimental results and simulation, there are several problems need to be solved before further consideration.

(1) The author strength the high device performance with the CuPc based VOFETs, which display a current density of 1 A/cm^2 but a limited current on/off ratio. However, there are literatures reported excellent electric properties with current density $> 1 \text{ A/cm}^2$ and high on/off ratio $> 10^6$ based on organic semiconductors, such as C8-BTBT and C60 (Nat. Commun. 2014, 5, 5162; Appl. Phys. Lett. 2004, 85, 5084.). The author should optimize parameters such as Aelect, OSC materials and film thickness, which confirm the effective enhancement of on current.

Author's response: We thank the reviewer for the fair evaluation of our manuscript. Following the suggestion of the reviewer, the reference corresponding to C₈-BTBT transistors (Nat. Commun. 2014, 5, 5162) has been inserted in the “comparison of J_D and I_{on}/I_{off} ” figure (Figure 9) of the revised manuscript as Ref. 74. The reference corresponding to C₆₀ devices (Appl. Phys. Lett. 2004, 85, 5084) was already included in the previous version of the manuscript (cited as Ref. 10 in the revised manuscript).

Although our values of on/off current ratio are limited, we have demonstrated high J_D values at ultra-low voltages ($\leq 3 \text{ V}$). The main goal of our manuscript is to demonstrate a novel

VOFET platform (based on a self-rolled drain electrode) that implies a source edge-driven working principle. This extends the operating concept of patterned-source VOFETs reported in the literature, and taking this into account we also investigate and explain the edge-driven concept in our manuscript using the theoretical modeling data. In addition, in the revised version of our manuscript, we have also tested the current response of the rolled-up NM VOFETs as a function of humidity and light exposure. The results show the multi-sensing capability of the VOFETs, which is unique compared to the devices reported in the literature.

Based on the comments of the reviewer, we have performed new experiments (described hereafter) in order to improve the on/off ratios of our VOFETs. After such experiments, the manuscript has been modified as well.

- (i) We have applied an irreversible compression over the tubular drain electrode, thus modifying its shape from cylindrical to a folded sheet. The irreversible compression was introduced by collapsing the tube electrode during the rolling-up process. As a consequence, the drain/semiconductor contacting area increased, as well as the on-off current ratio (to $\sim 10^2$). The new output characteristics are shown in Figure 6a, and the corresponding text has been inserted/modified on page 12 (highlighted in yellow). We have also included an optical microscopy image of a collapsed tube electrode as Supplementary Figure 2.
- (ii) We have increased the thickness of the Al_2O_3 gate dielectric layer (from 20 nm to 40 nm) to possibly tune its dielectric strength. As a consequence, we have obtained samples that support gate voltages up to 7 V. Under such conditions, the on-off current ratio also showed significant improvement to $\sim 10^2$. The new output characteristics are shown in Figure 6b, and the corresponding text has been inserted/modified on page 12 (highlighted in yellow).

(2) For the electric properties characterization, the author attributed the threshold voltage shifts and hysteresis to the absorption of water. As mentioned in this work, the measurements were performed in air, can the author compare the results with detection in N_2 conditions? What's more, there are reports via post-processing to reduce the solvent effect on organic electronic devices (Nat. Mater. 2017, 16, 356; Nat. Commun. 2019, 10, 2122; Adv. Mater. 2018, 30,

1801874.), the author need introduce post-treatment such as annealing, vacuum storage, to reduce the water effect.

Author's response: We thank the reviewer for their valuable suggestion regarding the reduction of hysteresis observed in our devices. Following the comments of the reviewer, we performed the following experiments to obtain maximum hysteresis control.

- (i) In order to reduce the damaging effects caused by the adsorption of water, we stored our samples in a vacuum for at least 3 days before the electrical measurements were performed.
- (ii) In our previous experiments, when the electrical characterization was performed using the “normal” configuration of the Semiconductor Parameter Analyzer, we realized that the current takes some time (tens of seconds) to reach the stationary state during the dc voltage sweep. Because of the current variations during the voltage sweep, the measurements performed using the “normal” configuration contributes to the hysteresis of the VOFETs. In order to resolve this, we performed experiments using the “quite” configuration of the Semiconductor Parameter Analyzer, which allows the current to reach the stationary state before the dc voltage sweep is initiated.

By performing the above-mentioned experiments, we observed a slight reduction in the hysteresis of our devices. The corresponding data is shown in Figure 5c in the revised manuscript while the corresponding text in the “Methods” section has also been added on page 23 (highlighted in yellow).

(3) The theoretical research did the series investigation on the thickness (50 nm to 3 μm) and source edges (26 to 68). The interesting thing is that the experiments were performed with the large channel length (9 and 3 μm). How about the device performance and carrier transport of thinner films devices?

Author's response: In principle, we performed our theoretical simulations considering three different source perforation gaps (50 nm, 200 nm, and 3 μm). For perforation gaps below 250 nm, J_D shows dependence on the perforation gap, as previously reported by Ben-Sasson et al. for patterned source VOFETs (*J. Appl. Phys.* **110**, 044501, 2011). On the other hand, upon reaching the threshold width of the source perforation gap (~250 nm), J_D starts to depend on the edges of

the source-electrode. In this scenario, an increase in the number of source edges is an important parameter to obtain a further increase in J_D . Indeed we demonstrated this in our experiments where 3 μm and 9 μm source perforation gaps were utilized. It was observed that an increase in the number of source edges from 26 to 68 (while keeping the source width constant) resulted in an increase in the current density by a factor of ~ 40 (Figure 5c of the manuscript). We find the comment of the reviewer very interesting as to what would be the effect of utilizing source perforation gaps narrower than 3 μm . Currently, we were able to obtain a resolution of 3 μm using the microfabrication techniques available at the Brazilian Nanotechnology National Laboratory. However, further improvement in this resolution is expected in the near future with the help of nano-lithography tools. As already discussed in our manuscript, the use of nano-lithography tools would allow the formation of a source-electrode consisting of a significantly large number of source edges. For instance, the formation of perforation gaps as narrow as 250 nm can yield 960 source edges (considering 240 μm wide source-electrode). In such a case, current densities of $\sim 10 \text{ A/cm}^2$ ($J_D/V_{GS} \approx 3 \text{ A/V.cm}^2$) can be achieved, assuming a linear relationship between the source edges and J_D . The aforementioned is discussed in detail in the “Theoretical modeling” and “Discussion” sections of the manuscript. Similarly, Figure 9 of the revised manuscript also shows the edge-driven prospective device which consists of 960 source edges and shows an estimated J_D/V_{GS} of $\sim 3 \text{ A/V.cm}^2$. The edge-driven operating mechanism explained in our manuscript extends the current understanding of VOFETs. The important point is that the edge-driven mechanism can be applied to drastically improve the J_D of both academically and industrially manufactured patterned-source VOFETs. The above discussion has now also been inserted in the “Introduction” and “Conclusions” sections of the revised manuscript (highlighted in yellow).

(4) The author demonstrated an effective strategy on VOFETs fabrication and the comparable properties to different OSCs (Figure 7). The author still needs to do more literature research to add materials with high J_D and $I_{\text{on}}/I_{\text{off}}$ (such as Nat. Commun. 2014, 5, 5162). What's more, can the author show us an application with these CuPc based VOFETs, which possess high current density at low-operation voltage? Such as in the phototransistor or memory elements (Nat. Photonics 2016, 10, 129; Adv. Mater. 2018, 30, 18036557; Adv. Mater. 2017, 29, 1604769; Adv. Funct. Mater. 2019, 29, 1808453.)

Author's response: We thank the reviewer for indicating the important reference (*Nat. Commun.* 2014, 5, 5162) which has now been included in the “comparison of J_D and I_{on}/I_{off} ” figure (Figure 9 of the revised manuscript) as Ref. 74.

Following the important suggestion of the reviewer, we have tested our VOFETs as humidity and light sensors. The use of rolled-up nanomembrane as the top drain electrode allows direct interaction between the semiconducting layer and external chemical analytes and light. Hence, evident changes in the magnitude and shape of the output characteristics were observed when the devices were exposed to incident light and increased relative humidity. In the case of phototransistors, the devices demonstrated responsivity and detectivity of ~ 0.05 A/W and $\sim 10^9$ Jones, respectively. The corresponding output characteristics are presented in the revised manuscript as Figure 7, while the corresponding discussion is inserted on pages 13 and 14. Our new multi-sensing results indicate that rolled-up nanomembrane VOFETs may also be used for the detection of various gases and biomolecules. Hence, our device platform should not only be seen as a transistor-alone platform but a potential candidate to develop and improve innovative sensing technologies.

Reviewers' comments:

Reviewer #1 (Remarks to the Author):

The researchers demonstrated a vertically stacked and individually controlled RGB OLED driven by TFT consisting of 2 transistor-1 capacitor model (2T-1C). They achieved a high aperture ratio of ~63%, which was made possible by employing photolithographic technique to fabricate intermediate electrodes between the individual stack. The driving transistor was connected to individual OLED unit through a via hole to each anode which was fabricated photolithographically as well. To facilitate the use of photolithographic technique the R, G and the B stack were separated from each other using an Al₂O₃/SiN_x where the Al₂O₃ layer served as thin film encapsulation to prevent degradation from moisture and oxygen and the SiN_x as a passivation layer to prevent degradation from solvents and other chemical involved during the fabrication. This particular combination of materials to avoid degradation of the OLED during processing were chosen for their low temperature fabrication and compatibility with photolithography. As for the electrical performance of the device, it exhibited a luminous current efficiency of 1.9 cd/A, 22.9 cd/A and 3.5 cd/A at 1000 cd/m² luminance and turn on voltage of 2.3V, 2.3V and 2.6V for the R, G and B unit respectively and covered a 113% sRGB color gamut. In the TFT configuration the data line required 5.9V for the R, 2.2V for the G and 6.3V for the B unit.

The researchers used photolithographic for fabricating intermediate electrodes and the via hole for TFT connection to the anode, but since OLED is susceptible to degradation from the photolithography process, they used intermediate layers of Al₂O₃/SiN_x which served as encapsulation and passivation layer which makes the device unique but also adds to the overhead time of making a device. It was also noted by the researchers that the electrical device performance wasn't affected by these additional layers (what about optical performance?)

The researchers did a good job in addressing the current limitation of TFT driven OLED which results in low aperture ratio and hence in low display resolution but then, this is also more of an engineering challenge than a scientific one.

I would like the researchers to address the following in their manuscript:

- Energy diagram of the intermediate layers to corroborate there is no leakage in between the OLED units
- Fig 1d is shown as a full color EL spectrum, which in my opinion is blue.
- Optical properties of the intermediate layer needs to be characterized, like information regarding the refractive indices and the transmissivity data for the Al₂O₃ layer is required at the least
- Specify the IZO etchant, PR stripper used in the photolithography process to make the results reproducible

Reviewer #2 (Remarks to the Author):

In the revised manuscript, the authors have addressed my comments and questions, so I recommend publication in Nature Communications.

Two more comments:

The comparison between the organic-transistor chip and the individually packaged MOSFET in Figure 2b makes no sense. In the text, the authors claim that the "conventional MOSFET is ... almost 20 mm in length." This is not true. The MOSFET has dimensions on the order of approximately 1 micron. It is true that the plastic housing has a size of 20 mm, but the only reason for making the package so large is to make it easier for the customers to hold the device in their hands and to be able to handle it, for example, to solder it onto a printed circuit board. The transistor itself is so small that you wouldn't be able to see it. The comparison in Figure 2b therefore makes no sense; you are comparing apples and oranges.

In Figure 5c, the authors need to show arrows to indicate the direction of the hysteresis.

Reviewer #3 (Remarks to the Author):

The authors have made sufficient modifications according to the modification comments, and I suggest that this paper be accepted without further modification.

Comments of Reviewer # 1 and response by the authors

- *In italic text*: the reviewer's comments.
 - In normal text: actions taken
-

Reviewer # 1:

The researchers demonstrated a vertically stacked and individually controlled RGB OLED driven by TFT consisting of 2 transistor-1 capacitor model (2T-1C). They achieved a high aperture ratio of ~63%, which was made possible by employing photolithographic technique to fabricate intermediate electrodes between the individual stack. The driving transistor was connected to individual OLED unit through a via hole to each anode which was fabricated photolithographically as well. To facilitate the use of photolithographic technique the R, G and the B stack were separated from each other using an Al₂O₃/SiN_x where the Al₂O₃ layer served as thin film encapsulation to prevent degradation from moisture and oxygen and the SiN_x as a passivation layer to prevent degradation from solvents and other chemical involved during the fabrication. This particular combination of materials to avoid degradation of the OLED during processing were chosen for their low temperature fabrication and compatibility with photolithography. As for the electrical performance of the device, it exhibited a luminous current efficiency of 1.9 cd/A, 22.9 cd/A and 3.5 cd/A at 1000 cd/m² luminance and turn on voltage of 2.3V, 2.3V and 2.6V for the R, G and B unit respectively and covered a 113% sRGB color gamut. In the TFT configuration the data line required 5.9V for the R, 2.2V for the G and 6.3V for the B unit.

The researchers used photolithographic for fabricating intermediate electrodes and the via hole for TFT connection to the anode, but since OLED is susceptible to degradation from the photolithography process, they used intermediate layers of Al₂O₃/SiN_x which served as encapsulation and passivation layer which makes the device unique but also adds to the overhead time of making a device. It was also noted by the researchers that the electrical device performance wasn't affected by these additional layers (what about optical performance?)

Author's response: We thank the reviewer for the fair evaluation of our manuscript. Though, there seems to be some misunderstanding because, in this contribution, we have shown that our transistors have the capability to detect humidity and light. On the other hand, the transistor-driven OLED setup (as mentioned by the reviewer) was not developed in this manuscript. However, we acknowledge that such a setup could be interesting and can be considered for a future contribution. In our manuscript, we report on the general working principle of VOFETs based on rolled-up nanomembranes and demonstrate their multi-sensing capability. Considering the typical stacking of electrodes, dielectric layer, and organic semiconductor in a vertical transistor, we have employed a patterned source that has enabled control over the VOFET current density by changing the number or size of source perforations (preserving the source porosity). Accordingly, the VOFET current density is found to be dependent on the number of source-edges. To complete the novelty of the device architecture, a nanomembrane-based drain is employed, enabling the fabrication of nanometer-thick vertical semiconducting channels. Such characteristics extend the concept of VOFETs operation. Finally, we have demonstrated that our novel VOFET is a suitable platform to develop sensor applications, such as light detection. In this case, a slight increment of drain currents is observed when the VOFET is illuminated (Figure 7b). We attribute this to the formation of photogenerated electron-hole pairs in the semiconducting channel, following the absorption of photons by the CuPc film. Regarding the reviewer's question about the optical performance of our VOFET devices, we have calculated the photoresponsivity (*ca.* 0.05 A/W) and the specific detectivity (*ca.* 10^9 Jones) – which are both typical parameters required to characterize the performance of phototransistors.

The researchers did a good job in addressing the current limitation of TFT driven OLED which results in low aperture ratio and hence in low display resolution but then, this is also more of an engineering challenge than a scientific one.

I would like the researchers to address the following in their manuscript:

- 1. Energy diagram of the intermediate layers to corroborate there is no leakage in between the OLED units*

Author's response: In our devices, two types of leakage currents are expected to flow. The first case corresponds to the gate leakage current (I_G), *i.e.* the current flowing through the gate

dielectric (Al_2O_3) layer between the source and gate electrodes. The I_G of our devices was recorded to be negligible (in the range of 10^{-12} ; as shown in Figure 4a). This is attributed to the preparation of a high-quality Al_2O_3 layer by atomic layer deposition, which effectively suppresses the leakage current between the source and gate electrodes. The low leakage current is also manifested by the overlap of J_D vs. V_{DS} curves at $V_{DS} = 0$ V (Figure 4b).

The second case corresponds to the leakage current between the source and drain electrodes (I_{off} i.e., I_D at $V_{GS} = 0$ V). This leakage takes place if the work function of the source metal is energetically close to the HOMO/LUMO of the used organic semiconductor. In our case, the source electrode is formed by 5 nm-thick Cr (work function: 4.5 eV), 10 nm-thick Au (work function: 5.1 eV) and 10 nm-thick SiO_2 . Although the work function of Au is close to the HOMO of CuPc (~ -5.2 eV), we observed a very low I_{off} of ~ 3 pA in our devices. The low values obtained for I_{off} are ensured by the deposition of a thin SiO_2 insulating layer on top of the source metal (as illustrated in Figure 1c), which partially shields the source electrode from the drain potential.

Following the suggestion of the reviewer, we have prepared an energy band diagram of the VOFET layers, which is included in the supplementary information as Supplementary Figure 2. Similarly, the main manuscript text was updated on page 8 (highlighted in yellow).

2. *Fig 1d is shown as a full color EL spectrum, which in my opinion is blue.*

Author's response: Figure 1d is actually a superposition of an optical image (colored) and a 2D monochromatic image. The monochromatic image corresponds to laser scanning (laser wavelength of 408 nm). The superposition of images is performed by the software of the Keyence VK-X200 3D laser scanning microscope, which acquires both images in the same measurement. This is the default configuration of the equipment. In order to elucidate our point, we have included some raw data files (Figure R1a,b) of the scanning, which result in the final image (Figure R1c) after superposition. It can be seen that the full color of Figure R1c is very similar to Figure 1d of the manuscript. We have now improved our description of Figure 1d to avoid misinterpretation (updated text is highlighted in yellow) and substituted the term “confocal microscopy” by “confocal laser scanning (CLS) microscopy” throughout the revised manuscript (also highlighted in yellow).

Figure R1: Micrographs of a VOFET with rolled-up drain electrode: (a) optical microscope image, (b) 2-dimensional monochromatic microscopy image, and (c) superposition of the optical and the monochromatic images, taken using the Keyence VK-X200 3D laser scanning microscope (default configuration).

3. *Optical properties of the intermediate layer needs to be characterized, like information regarding the refractive indices and the transmissivity data for the Al₂O₃ layer is required at the least*

Author's response: We understand the concern of the reviewer and we acknowledge that the optical properties of the gate dielectric layer can be considered important for the development of an OLED device. In our work, a VOFET device is reported, which is demonstrated to show the capability of humidity and light sensing. The first important point is that our VOFET device shows the capability of light-sensing and not light emission (as mentioned by the reviewer). Secondly, in the case of light-sensing, the VOFET devices were exposed to incident light from the side of the organic semiconductor and not from the gate dielectric side. Following the exposure to incident light, our devices showed a slight increase in drain current because the utilization of rolled-up NM electrode allows a large area of organic semiconductor to be exposed to the incident light thus enabling an effective absorption of photons by the organic semiconducting layer. The main role of the Al₂O₃ layer in our devices is to suppress the leakage current by electrically insulating the gate electrode from the source electrode. Nevertheless, to meet the requirements of the reviewer, we performed ellipsometry measurements on 70, 80, and 100 nm thick Al₂O₃ films, at the group of Professor A. A. Quivy (University of São Paulo (USP) – São Paulo, Brazil). In these measurements, a light source having a wavelength of 633 nm was used, providing a refractive index $n = 1.62 \pm 0.02$ for the above-mentioned thicknesses of Al₂O₃. This value is similar to that reported for thicker Al₂O₃ films grown by atomic layer deposition (ALD) on silicon substrates [Kumar, P. et al. 2009. doi: 10.1364/AO.48.005407]. The refractive index of 20–40 nm Al₂O₃ films (as used in our

VOFETs) has also been reported as *ca.* 1.6 [Shuzheng, S. et al. 2018. doi: 10.1155/2018/7598978]. Assuming the high reproducibility of Al₂O₃ thin films grown by ALD, we expect a transmittance of *ca.* 90-98% considering visible light, as also reported by Shuzheng, S. et al. using UV-VIS experiments. This agrees with the high Al₂O₃ films' transparency observed in our samples. We have now modified the manuscript accordingly, inserting the above results and discussions in the "Device microfabrication" section. The new text is highlighted in yellow.

4. *Specify the IZO etchant, PR stripper used in the photolithography process to make the results reproducible*

Author's response: We understand the concern of the reviewer. During the microfabrication of the rolled-up NM-based VOFETs, it was necessary to employ four different etching processes: (i) photoresist (PR) etching during lift-off processes, which is performed in acetone; (ii) SiO₂ etching to define the mesa structures, which is performed by reactive ion etching using CF₄ as the etching gas; (iii) Al₂O₃ etching to remove it from the top of gate contact pads, which is performed in hydrogen fluoride (HF) aqueous solution (1% v/v) for 18 s; and (iv) GeO_x etching to release the nanomembrane from the substrate, which is performed in H₂O₂ aqueous solution (0.25% v/v). The information regarding all the etchants can be found in the Methods section (now highlighted in yellow).

In the case of PR patterning processes, the photolithography masks were designed using a computer-aided program called CleWin. This information is provided in the caption of Figure 2 (now highlighted in yellow). The mask alignments and UV exposures were performed via direct-writing using a MicroWriter ML3 machine. The development after UV exposition was performed using AZ 726 MIF photoresist stripper. This information is provided in the Methods section (now highlighted in yellow).

Comments of Reviewer # 2 and response by the authors

- *In italic text*: the reviewer's comments.
 - In normal text: actions taken
-

Reviewer # 2:

In the revised manuscript, the authors have addressed my comments and questions, so I recommend publication in Nature Communications.

Two more comments:

1. The comparison between the organic-transistor chip and the individually packaged MOSFET in Figure 2b makes no sense. In the text, the authors claim that the “conventional MOSFET is ... almost 20 mm in length.” This is not true. The MOSFET has dimensions on the order of approximately 1 micron. It is true that the plastic housing has a size of 20 mm, but the only reason for making the package so large is to make it easier for the customers to hold the device in their hands and to be able to handle it, for example, to solder it onto a printed circuit board. The transistor itself is so small that you wouldn't be able to see it. The comparison in Figure 2b therefore makes no sense; you are comparing apples and oranges.

Author's response: We understand the concern of the reviewer and thus we have now removed the MOSFET photograph from Figure 2b. Correspondingly, the caption of Figure 2b has also been updated (highlighted in yellow).

2. In Figure 5c, the authors need to show arrows to indicate the direction of the hysteresis.

Author's response: In Figure 5c, arrows indicating the direction of hysteresis have now been included.

REVIEWERS' COMMENTS:

Reviewer #1 (Remarks to the Author):

The authors have made the changes and I recommend the manuscript for publication.